# Genomic malaria surveillance of antenatal care users detects reduced transmission following elimination interventions in Mozambique

Nanna Brokhattingen [1,7], Glória Matambisso[2,7], Clemente da Silva [2], Eric Neubauer Vickers[3], Arnau Pujol [1], Henriques Mbeve[2], Pau Cisteró [1], Sónia Maculuve[2], Boaventura Cuna[2], Cardoso Melembe[2], Nelo Ndimande[2], Brian Palmer[3], Manuel García-Ulloa[1], Humberto Munguambe[2], Júlia Montaña-Lopez [1], Lidia Nhamussua [2], Wilson Simone [2], Arlindo Chidimatembue[2], Beatriz Galatas[1,2], Caterina Guinovart [1], Eduard Rovira-Vallbona[1], Francisco Saúte[2], Pedro Aide [2], Andrés Aranda-Díaz[3], Bryan Greenhouse [3], Eusébio Macete[2,4] & Alfredo Mayor [1,2,5,6] ✉

Routine sampling of pregnant women at first antenatal care (ANC) visits could make *Plasmodium falciparum* genomic surveillance more cost-efficient and convenient in sub-Saharan Africa. We compare the genetic structure of parasite populations sampled from 289 first ANC users and 93 children from the community in Mozambique between 2015 and 2019. Samples are amplicon sequenced targeting 165 microhaplotypes and 15 drug resistance genes. Metrics of genetic diversity and relatedness, as well as the prevalence of drug resistance markers, are consistent between the two populations. In an area targeted for elimination, intra-host genetic diversity declines in both populations (p = 0.002-0.007), while for the ANC population, population genetic diversity is also lower (p = 0.0004), and genetic relatedness between infections is higher (p = 0.002) than control areas, indicating a recent reduction in the parasite population size. These results highlight the added value of genomic surveillance at ANC clinics to inform about changes in transmission beyond epidemiological data.

Recent innovations in sequencing technologies have created new opportunities for the strategic use of genomics in malaria surveillance[1–3]. Examples include more accurate data on emergence and spread of drug and diagnostic resistance[4,5], inferring parasite connectivity to support the classification of imported cases[6], and predicting vaccine effectiveness[7]. Furthermore, and still at a more theoretical stage, genomic diversity can be used to assess differences and changes in transmission intensity[8–12]. This could be especially

[1]ISGlobal, Hospital Clínic – Universitat de Barcelona, Barcelona, Spain. [2]Centro de Investigação em Saúde de Manhiça (CISM), Maputo, Mozambique. [3]EPPIcenter Research Program, Division of HIV, Infectious Diseases, and Global Medicine, Department of Medicine, University of California, San Francisco, California, USA. [4]National Directorate for Public Health, Ministry of Health, Maputo, Mozambique. [5]Spanish Consortium for Research in Epidemiology and Public Health (CIBERESP), Madrid, Spain. [6]Department of Physiological Sciences, Faculty of Medicine, Universidade Eduardo Mondlane, Maputo, Mozambique. [7]These authors contributed equally: Nanna Brokhattingen, Glória Matambisso. ✉e-mail: alfredo.mayor@isglobal.org

useful for stratification and evaluating the effectiveness of anti-malarial interventions.

For continuous genomic surveillance of malaria, samples must be collected regularly, and, especially critical for low-resource settings, cost-efficiently[2,13,14]. Pregnant women attending their first antenatal care (ANC) consultation are an easy-access subpopulation that could potentially serve as a sentinel group for malaria surveillance[13,15,16]. Besides low cost and easy accessibility, advantages of ANC-based surveillance include temporal continuity, known denominator populations, and the possibility of capturing asymptomatic infections[15]. Malaria burden trends in pregnant women at their first ANC visit have been shown to mirror community trends[17], and routine malaria testing at ANC has already been implemented in Tanzania, where it is generally perceived as acceptable and positive by both patients and providers[18]. A few small studies, mostly outside of Africa, have investigated malaria genetic diversity in pregnant women using whole genome sequencing[19], microsatellite markers[20,21] or nested polymerase chain reaction (PCR)[22,23]. However, routinely collected genomic data from ANC has not been evaluated for its suitability for sentinel surveillance.

We hypothesized that the *Plasmodium* (*P.*) *falciparum* parasite population circulating in pregnant women at their first ANC visit and in the community are genetically similar, including similar genetic diversity (intra-host and population-level), relatedness between infections, and prevalence of antimalarial resistance markers. To test our hypothesis, we analyzed the parasite population in ANC users in southern Mozambique, and compared it to parasites found in children aged 2–10 years sampled in household surveys. Furthermore, we compared the parasite populations in three areas with declining transmission between 2015 and 2018. Manhiça and Magude are low-transmission areas, with Magude recently targeted for elimination with a package interventions[24], while Ilha Josina is a historically high-transmission setting[17].

## Results

### Sequencing performance
A total of 558 *P. falciparum*-positive dried blood spot (DBS) samples from ANC users ($n = 378$) and children sampled in population-representative household surveys ($n = 180$) were sequenced.

We used a panel targeting 241 amplicons in the *P. falciparum* genome[25]. The amplicons included markers of drug resistance in 15 genes and 165 microhaplotypes that had been selected to provide information about genetic diversity in the parasite population. On average, quality reads were obtained for 212 and 170 loci per sample for ANC users and children, respectively (Table 1). 68.5% (382/558) of the samples passed the filtering criteria. A lower proportion of samples from children passed filtering (51.7%) compared to ANC users (76.5%). Total number of reads per sample and number of loci covered per sample ($n = 558$) was primarily a function of parasite density (Fig. 1A, B). Across all samples sequenced, parasite densities were lower in those from children than those from ANC (Table 1). Parasite densities were similar between populations among samples that passed filtering (geometric mean [GM] = 191 and GM = 154 parasites/μL, respectively), and among samples that were filtered out (GM = 7 parasites/μL for both, Fig. 1C). Sequencing coverage was high across samples that passed filtering ($n = 328$), with a geometric mean total reads per sample of 319,503 and 484,245, and a median 223 and 213 out of 241 loci covered per sample, for ANC users and children, respectively (Table 1). On average, each locus ($n = 241$) was covered by 1.4 million reads and reads from 462 samples (Fig. 1D, E).

### Intra-host genetic diversity
Half of the pregnant women attending ANC consultations carried polyclonal infections (Table 2). On average, ANC users had a multiplicity of infection (MOI; unadjusted for relatedness) of 2.37, i.e., carried 2.37 genetically different *P. falciparum* parasite clones

(Supplementary Table 1). 1-$F_{ws}$ was 0.39, suggestive of inbreeding in the population. Effective multiplicity of infection (eMOI), which incorporates intra-host relatedness between clones, was lower than MOI at 1.8, indicative of co-transmission over super-infections, leading to inbreeding (Supplementary Fig. 1C). Parasite density was associated with measured intra-host diversity, with higher diversity observed for women with higher-density infections. eMOI showed an overall declining trend from 2017 to 2018–2019, and was highest in Magude. 1-$F_{ws}$ showed similar trends but did not reach statistical significance. Primigravid women had higher eMOI compared to multigravidae in the univariate analysis, but the effect disappeared when adjusting for parasitemia, time, and area (Supplementary Table 2). No statistically significant differences were observed between seasons or human immunodeficient virus (HIV)-status groups. Among children, 62.4% carried polyclonal infections, the average MOI and eMOI was 2.86 and 2.3, respectively, and 1-$F_{ws}$ was 0.55. Similar to ANC users, children with higher-density infections showed higher eMOI.

### Temporal trends in intra-host genetic diversity
A significant interaction was observed between area and time in the multivariate analysis of intra-host diversity at ANC, indicating different temporal trends within the three areas. Parasite densities did not change over time (Supplementary Fig. 2). In Magude, eMOI declined by 50% per year (95%CI: −0.78;−0.25, $p < 0.0002$, Fig. 2A–C, Supplementary Table 3), with a shift toward more infections having eMOI > 2 (Supplementary Fig. 3), while 1-$F_{ws}$ and odds of infections being polyclonal showed declining trends (58% and 46% yearly decline, respectively). Similar trends were observed from naively estimated MOI and MOI not adjusted for intra-host relatedness (Supplementary Fig. 4). No temporal changes in intra-host diversity were observed in Manhiça, while in Ilha Josina, there was an increasing trend over time in polyclonal infections, although not statistically significant (Fig. 2B). Intra-host diversity among 47 children from Magude sampled cross-sectionally were compared with samples from ANC users in Magude (Fig. 2A–C Magude panel and D–F, Supplementary Table 4). In multivariate regressions combining both populations, all metrics of intra-host diversity showed declining trends over time. Both populations showed very significant declines in eMOI (−37% and −51% per year for children and ANC users, respectively), and eMOI was not associated with population group ($p = 0.20$). 1-$F_{ws}$ also tended to decline in both population groups, and no effect of population was detected. Further comparisons between ANC and children in the other two areas were precluded by limited number of samples from children.

### Relationship between intra-host genetic diversity and parasite rates
To assess the potential of using intra-host genetic diversity as a proxy for transmission intensity, we compared mean eMOI, proportion polyclonal, and mean 1-$F_{ws}$, with another proxy for transmission intensity, *P. falciparum* parasite rates using quantitative polymerase chain reaction (qPCR) (Fig. 2, Supplementary Figs. 5, 6). *P. falciparum* parasite rates in ANC users declined in all three areas during the study, starting from a higher rate in Ilha Josina compared to Magude and Manhiça (previously reported in[17]). In Magude, eMOI in both ANC users and children showed positive Pearson's correlation coefficients (PCC) close to 1 (>0.85), although not statistically significant. Furthermore, for ANC users in Magude, both 1-$F_{ws}$ and proportion of infections that were polyclonal showed PCC > 0.65. In the other two areas, PCC was negative, but small to moderate and not statistically significant, for all three metrics of intra-host genetic diversity.

### Population genetic diversity
Among ANC users, population mean expected heterozygosity ($H_E$) across the 165 microhaplotype loci ranged from <0.01 to 0.90, with a

**Table 1 | Characteristics of study participants by population group and inclusion in analysis after filtering**

| | First ANC users | | | Children 2–10 years | | |
|---|---|---|---|---|---|---|
| | All | Passed filtering | Filtered out | All | Passed filtering | Filtered out |
| Total number N (%) | 378 (100) | 289 (76.5) | 89 (23.5) | 180 (100) | 93 (51.7) | 87 (48.3) |
| Age median years (IQR) | 21 (18;27) | 21 (18;27) | 22 (19;29) | 4.0 (3.0;6.3) | 4.6 (3.0;6.8) | 4.0 (3.0;6.0) |
| Parasite density GM parasite/ μL (GSD) | 86.6 (16.9) | 191 (12.4) | 6.6 (9.0) | 41.1 (18.7) | 154 (12.8) | 7.1 (11.7) |
| Year | | | | | | |
| 2015–2016, n (%) | 0 | 0 | 0 | 93 (100) | 60 (64.5) | 33 (35.5) |
| 2017, n (%) | 210 (100) | 166 (79.0) | 44 (21.0) | 21 (100) | 4 (19.0) | 17 (81.0) |
| 2018–2019, n (%) | 168 (100) | 123 (73.2) | 45 (26.8) | 66 (100) | 29 (43.9) | 37 (56.1) |
| Area | | | | | | |
| Magude, n (%) | 155 (100) | 120 (77.4) | 35 (22.6) | 101 (100) | 47 (46.5) | 54 (53.5) |
| Manhiça, n (%) | 87 (100) | 64 (73.6) | 23 (26.4) | 71 (100) | 42 (59.2) | 29 (40.8) |
| Ilha Josina, n (%) | 136 (100) | 105 (77.2) | 31 (22.8) | 8 (100) | 4 (50.0) | 4 (50.0) |
| Season | | | | | | |
| Dry, n (%) | 233 (100) | 182 (78.1) | 51 (21.9) | 180 (100) | 93 (51.7) | 87 (48.3) |
| Rainy, n (%) | 145 (100) | 107 (73.8) | 38 (26.2) | 0 (0) | 0 (0) | 0 (0) |
| HIV | | | | | | |
| Negative, n (%) | 291 (100) | 221 (75.9) | 70 (24.1) | NA | NA | NA |
| Positive, n (%) | 87 (100) | 68 (78.2) | 19 (21.8) | NA | NA | NA |
| Gravidity | | | | | | |
| Primigravid, n (%) | 143 (100) | 117 (81.9) | 26 (18.2) | NA | NA | NA |
| Multigravid, n (%) | 235 (100) | 172 (73.2) | 63 (26.8) | NA | NA | NA |
| Sequencing coverage | | | | | | |
| Reads/sample, GM (GSD) | $6.8 \times 10^4$ (27.2) | $3.2 \times 10^5$ (4.0) | 470 (15.5) | $8.5 \times 10^3$ (100.0) | $4.8 \times 10^5$ (5.3) | 115 (8.7) |
| N loci* covered/sample, median (IQR) | 221 (173;224) | 223 (209;225) | 75 (25;128) | 170 (41;214) | 213 (208;224) | 40 (13.5;81) |

All available samples were sequenced. Those that did not pass the filtering criteria were excluded, while the remaining samples were included in the analysis. Population groups are (1) pregnant women attending their first antenatal care (ANC) visit, and (2) children aged 2-10 years sampled in population-representative household surveys.
N number, IQR interquartile range, GM geometric mean, GSD geometric standard deviation, NA not applicable. *On average, each locus was covered by 1.4 million reads and 462 samples.

mean of 0.57 (95% CI: 0.54–0.60, Fig. 3A). Three to 58 unique alleles were observed for each locus (Fig. 3B). In order to compare $H_E$ between time windows, areas, and population groups, the larger populations were randomly subsampled within areas and/or years once without replacement to match the smaller population in size. Overall, $H_E$ did not change between 2017 and 2018–2019 (Fig. 3E, F, Supplementary Table 7). Comparing $H_E$ between ANC populations in the three areas, parasites in Magude showed less diversity than the parasite population in Ilha Josina (Fig. 3C, D, Supplementary Table 7). Mean $H_E$ did not differ between ANC and children populations ($p = 0.95$, Fig. 3G, H, Supplementary Table 8).

**Pairwise inter-host genetic relatedness**

Genetic relatedness between pairs of *P. falciparum* infections, including unphased polyclonal infections, was estimated with an identity by descent (IBD)-based approach. To compare relatedness between areas and populations, we performed permutation of labels and compared mean IBD in each area or population with permutation distributions. Infections from ANC users ($n = 83,521$ pairs) generally showed low relatedness, with a mean pairwise IBD of 0.026 (95% CI: 0.022;0.033) (Supplementary Fig. 7). IBD was slightly but significantly higher between infections in Magude compared to within and between other areas (Supplementary Fig. 7a, Supplementary Table 9). Infections in children tended to be more related compared to infections in ANC users, and between the two populations. Restricting the comparison to samples from overlapping years (2017–2020) and temporal windows (April 15 to June 30), mean IBD between ANC infections was 0.018, similar to the mean IBD of 0.017 observed for infections in children (Supplementary Chary Fig. 7b, Supplementary Table 10).

**Markers of drug resistance**

The prevalence of all markers of antimalarial resistance assessed in this study were similar between ANC users and children from the community (Table 3). Parasites with quintuple 51-59-108-437-540 mutations in the *dihydrofolate reductase* and *dihydropteroate synthetase* (*pfdhfr-pfdhps*) genes were highly prevalent in both populations (>90%). In particular, sulphadoxine-pyrimethamine (SP) resistance-associated polymorphisms in the *pfdhfr* gene had almost reached fixation in the population, with 98.6% carrying the triple 51-59-108 mutant. No A581G nor I431V mutations in *pfdhps* were detected. Three quarters of the study population carried a *multidrug resistance 1* (*pfmdr1*) F184Y gene mutation associated with amodiaquine resistance, while 1.2% carried the N86Y, and 0.3% carried the D1246Y mutations. The *chloroquine resistance transporter* (*pfcrt*) 72-76 CVIET mutant genotype was observed in four individuals, three of them children. No mutations in the *kelch 13 propeller* gene (*pfkelch13*) associated with artemisinin partial resistance, was observed in either population.

## Discussion

This study applied a multiplexed amplicon sequencing approach targeting microhaplotypes and drug resistance markers to assess the representability of pregnant women attending their first ANC consultation for sentinel *P. falciparum* genomic surveillance. We found that genetic diversity and pairwise inter-host relatedness, as well as prevalence of drug resistance markers, were consistent between first ANC users and children aged 2–10 years, representing the community. In Magude, which was subject to an elimination campaign, similar declining trends in intra-host diversity were observed for both ANC

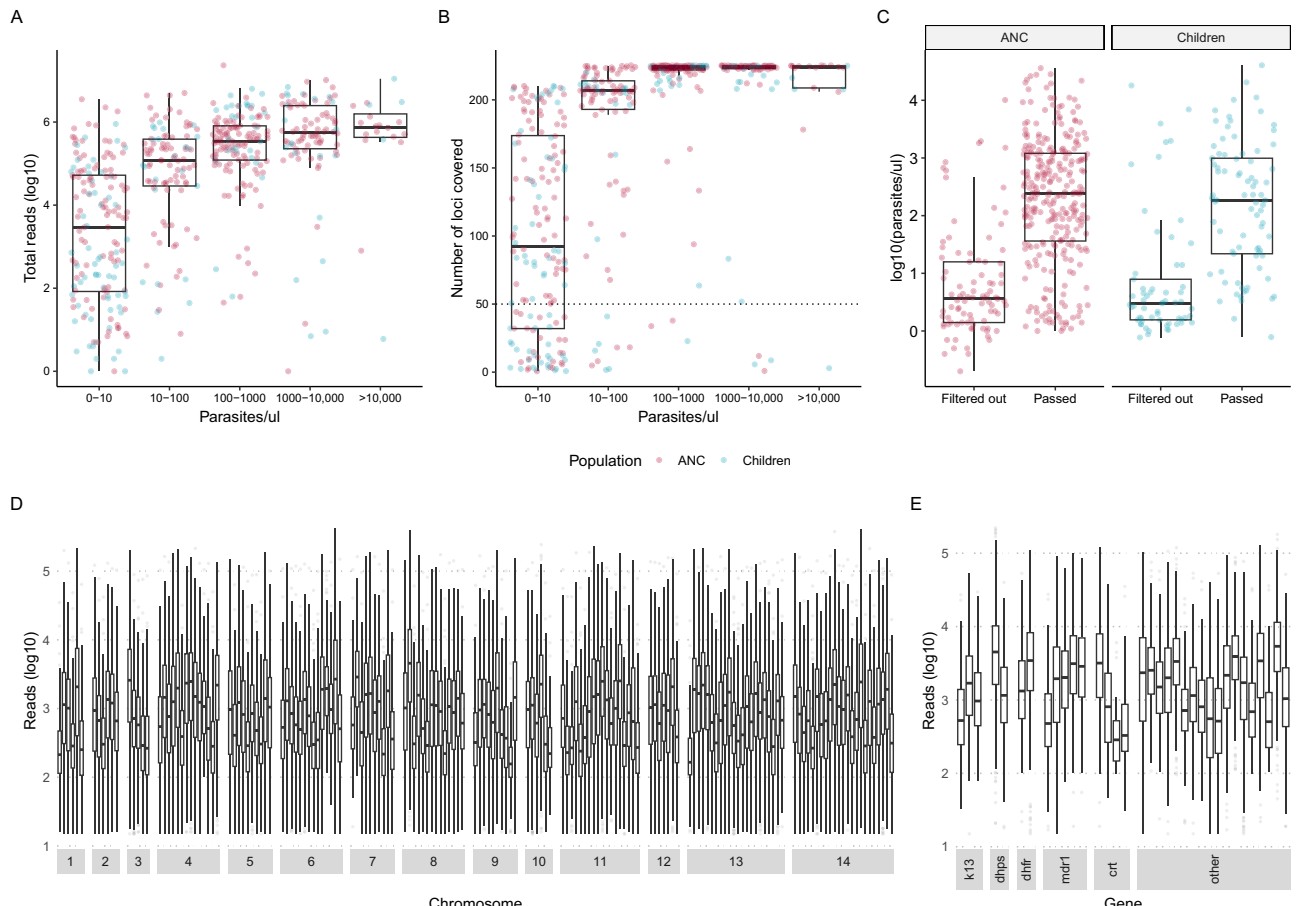

**Fig. 1 | Sequencing performance. A** Total number of reads per sample (*n* = 558) by parasite density before filtering. Red indicates pregnant women at their first ANC visit, and blue indicates children sampled in household surveys. **B** Number of loci (total *n* = 224) with reads per sample (*n* = 558) by parasite density before filtering. **C** Parasite density among samples that passed and did not pass filtering by population group (*n* = 378 ANC users and *n* = 180 children). **D, E** Reads per locus per included sample (*n* = 382) for diversity loci by chromosome and for drug resistance markers by gene, respectively. On average, each locus was covered by 1.4 million reads. Boxes indicate the 25th and 75th percentiles with the centre line indicating the median, and the whiskers indicate the smallest value within 1.5 times inter-quartile range below the 25th percentile, and the largest value within 1.5 times the interquartile range above 75th percentile.

users and children. Our findings demonstrate the potential of ANC-based malaria genomics as a straight-forward and cost-efficient approach to assess the impact of antimalarial interventions and genetic variants of public health concern.

Pregnant women seeking ANC have previously been shown to mirror trends in malaria prevalence in the general population, although with a delay, and with more heterogeneity between gravidity groups at higher transmission settings[16,17]. A few studies have also compared the genetic diversity of parasite populations in pregnant women and the community[19,20,22,23], but these were based on small sample sizes, only one took place in Africa[23], and, importantly, none accounted for parasite densities. With this study, we expand the potential scope of ANC-based surveillance to include genomic surveillance of *P. falciparum* genetic diversity and resistance markers. We find that both primigravid and multigravid first ANC users, regardless of HIV status, can be included in a sentinel population. Since no difference was observed between seasons, sampling could take place throughout the year. However, other studies did find seasonal differences[8], indicating that this might depend on the setting. Furthermore, it may not be realistic to reach sufficient sample sizes at ANC facilities alone at very low transmission, and it would be necessary to combine ANC sampling with other sampling strategies, such as health facility surveys. ANC sampling would also not be ideal if the goal is to identify finer relatedness patterns, including transmission networks, because of the temporal sparsity of samples. The very low inter-host relatedness observed at ANC might reflect little localized transmission, although more dense sampling would probably be required to detect this. Consistent with previous observations that parasite populations are at least partially structured in time[26], inter-host relatedness was higher among cross-sectionally sampled children than among continuously sampled ANC users, with the difference disappearing when restricting the comparison to similar temporal windows.

Genetic diversity has been proposed as a surrogate marker of transmission intensity[4,9,10,12]. In line with this and previous studies[8,11], we found the highest population diversity ($H_E$) in the highest-transmission setting, Ilha Josina. Conversely, we also found the lowest intra-host diversity in Ilha Josina (both eMOI and 1-$F_{ws}$). This might be explained by importation of parasites to low-transmission Magude and Manhiça from areas with higher transmission. A study from nearby low-transmission Eswatini observed similarly high diversity, which was attributed to frequent importation[27]. Alternatively, pregnant women in Iha Josina were previously found to, on average, have lower parasite densities compared to women from Magude and Manhiça[28], probably reflecting higher levels of anti-parasite immunity. Since parasite density was a major predictor of intra-host diversity, this could affect the likelihood of encountering and being able to measure multiple clones in these women. Comparing *P. falciparum* parasite rates with metrics of intra-host genetic diversity, we only observed positive correlations in Magude. This indicates that other factors might affect the relationship between parasite rates and genetic diversity, such as the temporal

**Table 2 | Factors associated with intra-host *Plasmodium falciparum* diversity among first antenatal care users and children**

| First ANC users | eMOI (95% CI) | p | p* | % polyclonal (95% CI) | p | p* | 1-$F_{ws}$ (95% CI) | p | p* |
|---|---|---|---|---|---|---|---|---|---|
| All (n = 289) | 1.82 (1.72;1.93) | | | 51.2 (45.5;56.9) | | | 0.39 (0.30;0.50) | | |
| Year | | 0.37 | 0.023 | | 0.82 | 0.18 | | 0.65 | 0.39 |
| 2017 (n = 166) | 1.86 (1.73;2.01) | | | 50.6 (42.8;58.4) | | | 0.41 (0.29;0.57) | | |
| 2018–2019 (n = 123) | 1.76 (1.50;2.16) | | | 52.0 (42.9;61.1) | | | 0.36 (0.15;0.85) | | |
| Change per year | −0.14 (−0.31;0.03) | 0.10 | 0.0010 | NA | NA | 0.040 | −0.10 (−0.44;0.23) | 0.56 | 0.41 |
| Area | | 0.015 | 0.00083 | | 0.71 | 0.21 | | 0.36 | 0.27 |
| Magude (n = 120) | 2.00 (1.83;2.19) | | | 53.3 (44.0;62.4) | | | 0.48 (0.32;0.70) | | |
| Manhiça (n = 64) | 1.73 (1.43;2.21) | | | 46.9 (34.5;59.7) | | | 0.33 (0.11;0.95) | | |
| Ilha Josina (n = 105) | 1.67 (1.41;2.06) | | | 51.4 (41.5;61.2) | | | 0.33 (0.12;0.85) | | |
| Parasite density | | 1.9e-08 | 3.0e-09 | | 0.0011 | 0.00041 | | 0.11 | 0.12 |
| <100 p/μL (n = 103) | 1.44 (1.32;1.58) | | | 35.9 (26.9;46.0) | | | 0.26 (0.16;0.42) | | |
| 100–<1000 p/μL (n = 106) | 1.89 (1.46;2.69) | | | 62.3 (52.3;71.3) | | | 0.44 (0.14;1.30) | | |
| ≥1000 p/μL (n = 80) | 2.21 (1.63;3.30) | | | 56.2 (44.7;67.2) | | | 0.51 (0.19;1.59) | | |
| HIV | | 0.15 | 0.27 | | 0.27 | 0.28 | | 0.42 | 0.46 |
| Negative (n = 220) | 1.78 (1.67;1.90) | | | 49.3 (42.6;56.1) | | | 0.37 (0.27;0.49) | | |
| Positive (n = 69) | 1.96 (1.61;2.47) | | | 57.4 (44.8;69.1) | | | 0.47 (0.19;1.12) | | |
| Gravidity | | 0.0062 | 0.20 | | 0.47 | 0.75 | | 0.39 | 0.71 |
| Primigravid (n = 117) | 2.00 (1.83;2.19) | | | 53.8 (44.4;63.0) | | | 0.44 (0.30;0.65) | | |
| Multigravid (n = 172) | 1.70 (1.45; 2.07) | | | 49.4 (41.8;57.1) | | | 0.35 (0.14;0.88) | | |
| Season | | 0.46 | 0.73 | | 0.63 | 0.38 | | 0.98 | 0.89 |
| Dry (n = 182) | 1.85 (1.72;1.99) | | | 50.0 (42.8;57.2) | | | 0.39 (0.28;0.53) | | |
| Rainy (n = 107) | 1.77 (1.50;2.17) | | | 53.3 (43.4;62.9) | | | 0.39 (0.16;0.90) | | |
| **Children 2–10 years** | | | | | | | | | |
| All (n = 93) | 2.30 (2.08;2.54) | | | 62.4 (52.2;71,5) | | | 0.55 (0.36;0.84) | | |
| Year | | 6.0e-06 | 0.00022 | | 0.81 | 0.94 | | 0.41 | 0.71 |
| 2015–2016 (n = 60) | 2.65 (2.35;3.00) | | | 63.3 (49.8;75.1) | | | 0.68 (0.40;1.12) | | |
| 2017 (n = 4) | 1.96 (1.23;3.81) | | | 75.0 (21.9;98.7) | | | 0.49 (0.02;6.52) | | |
| 2018–2019 (n = 29) | 1.60 (1.28;2.17) | | | 58.6 (39.1;75.9) | | | 0.35 (0.02;1.51) | | |
| Change per year | −0.41 (−0,59;−0.25) | 2.0e-07 | 0.00017 | NA | NA | 0.95 | −0.21 (−0.55;0.12) | 0.22 | 0.77 |
| Area | | 0.0052 | 0.0013 | | 0.51 | 0.082 | | 0.25 | 0.58 |
| Magude (n = 47) | 2.40 (2.09;2.76) | | | 61.7 (46.4;75.1) | | | 0.52 (0.28;0.94) | | |
| Manhiça (n = 42) | 2.30 (1.71;3.35) | | | 69.0 (52.8;81.9) | | | 0.66 (0.15;2.85) | | |
| I. Josina (n = 4) | 1.00 (-Inf;1.53) | | | NA* | | | 0.04 (NA;1.74) | | |
| Parasite density | | 3.0e-07 | 8.0e-07 | | 0.73 | 0.48 | | 0.53 | 0.46 |
| <100 p/μL (n = 33) | 1.76 (1.50;2.10) | | | 60.6 (42.2;76.7) | | | 0.42 (0.19;0.86) | | |
| 100–<1000 p/μL (n = 28) | 2.43 (1.57;4.43) | | | 67.9 (47.6;83.4) | | | 0.65 (0.10;3.93) | | |
| ≥1000 p/μL (n = 20) | 3.29 (1.93;6.45) | | | 70.0 (45.7;87.2) | | | 0.78 (0.11;5.17) | | |

*eMOI* effective multiplicity of infection, estimated with MOIRE (R package). Means and 95% confidence intervals (CI) obtained from 0-truncated Poisson regression, with *p* values computed with likelihood ratio tests.

Polyclonal infections are defined as having an eMOI>1.1. 95% CI obtained with *Z* test of proportions, *p* values with chi square goodness-of-fit test, and adjusted *p* values with likelihood ratio test in multivariate logistic regression.

$F_{ws}$ within-host heterozygosity relative to population-level expected heterozygosity at a given allele. Means and 95% CI are obtained from logistic regression, with *p* values from likelihood ratio tests.

*p* value for univariate analysis, *p** for multivariate analysis adjusting for parasitemia (categorical), time (continuous), area, and an interaction between time and area, *NA* not applicable.

*No polyclonal infections were detected among children in Ilha Josina (0/4).

scales at which malaria transmission and genetic diversity change[10,29]. Furthermore, the sites are likely to differ in host immunity due to previous exposure[28] which, again, might affect the chance of measuring multiple clones in one individual. Similarly, healthcare-seeking behavior and the use of antimalarials might differ between the sites. Finally, it cannot be discarded that diversity statistics may be more directly impacted by the interventions deployed in Magude, with the transmission decline being a correlation. Genetic diversity on its own might, therefore, not always be a suitable proxy for local transmission intensity, and stratification based on genetic metrics should be

carefully validated against other epidemiological data, including assessing the potential role of importation and underlying reasons for changes to transmission intensity.

On the other hand, genetic indicators of reduced transmission observed within Magude (decline in eMOI and 1-$F_{ws}$, increased mean IBD, and lower $H_E$) highlight how parasite genomics can complement clinical and epidemiological data to evaluate the impact of control interventions. Between 2015 and 2017, Magude was targeted with biannual rounds of mass drug administration (MDA), followed by reactive focal MDA in 2018, and three rounds of indoor residual

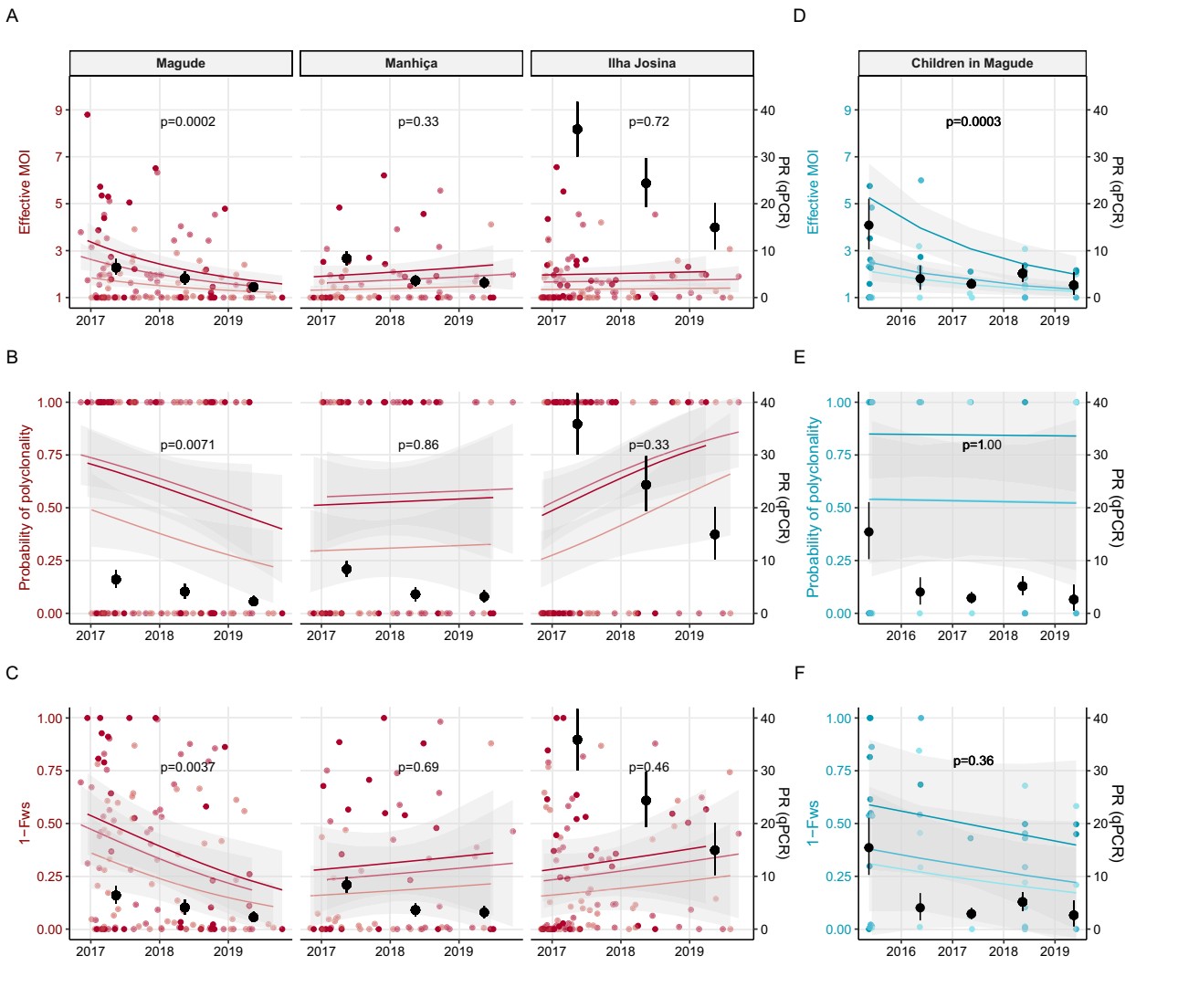

**Fig. 2 | Temporal trends in *Plasmodium falciparum* intra-host genetic diversity among first antenatal care users by area and children in Magude.** Intra-host genetic diversity over time in pregnant women attending their first antenatal care (ANC) visit (in red) by area (*N* = 120 in Magude, *n* = 64 in Manhiça, *n* = 105 in Ilha Josina), and children aged 2–10 years old from the community (blue, *n* = 47). Darker shade of color reflects higher parasite densitiy. Black dots represent *Plasmodium falciparum* (*Pf*) parasite rates (PR) by qPCR in the same population with 95% confidence intervals (CI) bars. **A** Effective multiplicity of infection (eMOI) in first ANC users by area in 0-truncated Poisson regression adjusted for parasitemia (*Pf* parasites/μl) with 95% CI bands. *P* values for temporal trend of eMOI in the regression (two-sided test) adjusted for multiple testing. **B** Monoclonal (eMOI ≤ 1.1) and polyclonal (eMOI>1.1) infections in pregnant women at ANC by area in a logistic regression adjusted for parasite density with 95% CI bands. **C** 1-$F_{ws}$ in pregnant women at ANC by area in a logistic regression adjusted for parasitemia with 95% CI bands. **D–F** eMOI, polyclonality and 1-$F_{ws}$ in children from Magude, estimated with Poisson and logistic regressions similar to **A–C** with 95% CI bands. *P* values in all graphs are for the temporal trend of the given metric in the regression (*F* test). Adjusted for multiple testing using the Benjamin–Hochberg method, a *p* value of <0.0062 indicates statistical significance.

spraying (IRS)[24]. Even though parasite rates declined in all three areas during the study period, and at similar levels and rates in Magude and the control area Manhiça[28], we only observed evidence of declining intra-host diversity in Magude. Furthermore, Magude showed significantly lower population diversity and higher mean IBD compared to the other areas. A study from Zambia found a similar reduction in the complexity of infections following an MDA trial[30]. These findings reveal programmatically important changes to the parasite population structure, not apparent from prevalence and incidence estimates.

Strengths of this study include the rich data obtained from deep amplicon sequencing, with sensitivity to achieve good coverage for samples with down to 10 parasites/μL. A previous study comparing amplicon sequencing to whole genome sequencing showed that amplicon sequencing provides higher coverage, thereby allowing for more sensitive detection of minority strains, even for infections with high parasite densities[31]. Compared to single nucleotide polymorphism (SNP)-based methods, microhaplotypes allow for higher resolution and consequently more accurate estimates of diversity and relatedness, while being more convenient than microsatellites[9,31]. Microhaplotypes have previously been used in a nation-wide study from Mozambique, where they proved able to distinguish parasites from the northern and southern parts of the country[4]. Furthermore, although possible to distinguish major and minor alleles in polyclonal infections using SNPs, highly diverse microhaplotypes allow a more accurate assessment and better utilization of the full allele diversity in polyclonal samples[31], which was half of the samples in this study. Relatedness between clones within a sample was evident from eMOI being lower than MOI, indicating that co-transmission is a more

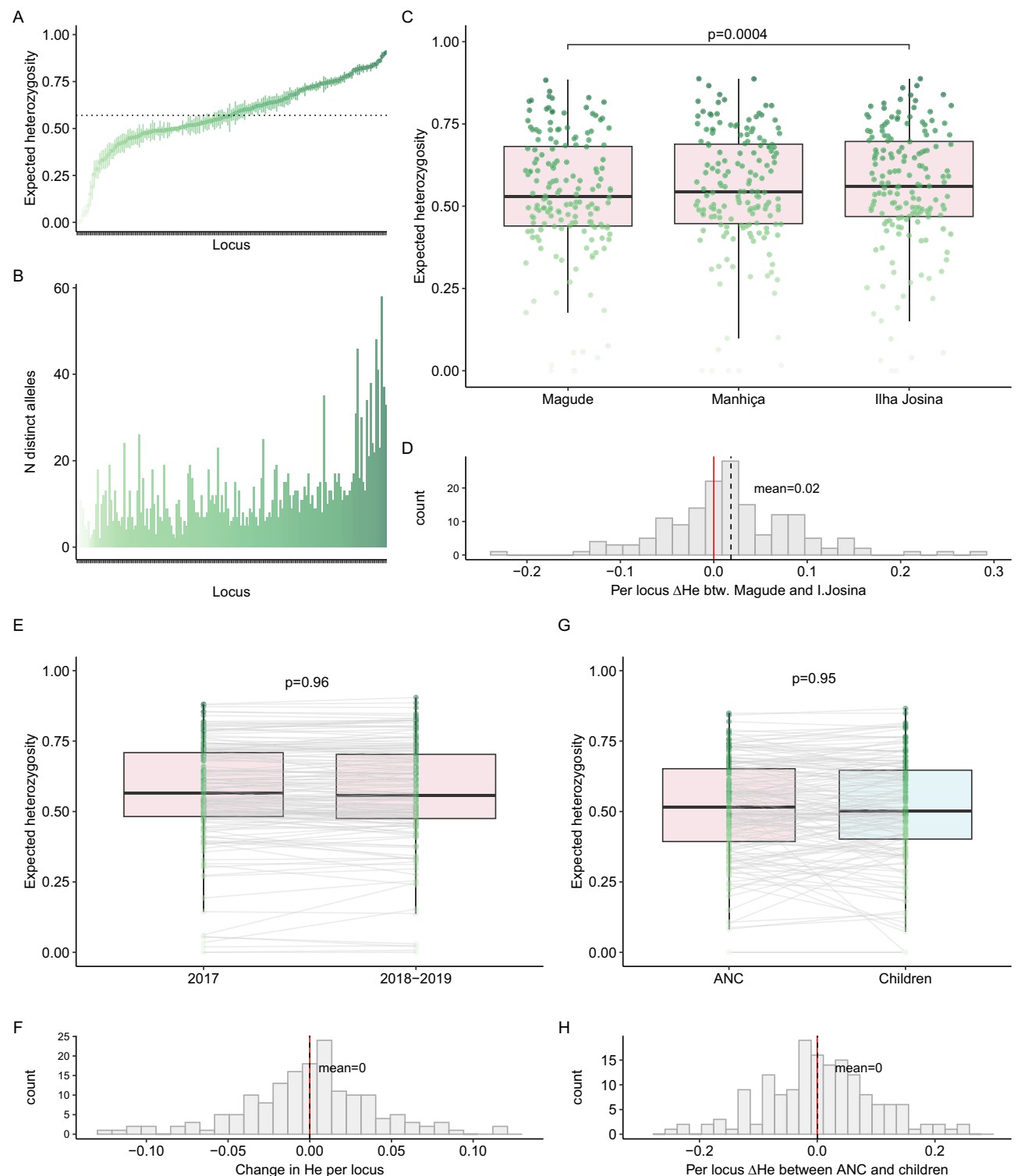

common event than superinfections. For our main analysis, we take this relatedness into account by using eMOI rather than MOI to estimate within-host diversity. Another strength of this study is the large ANC sample size, collected prospectively across three years in three different transmission scenarios. To the best of our knowledge, this study represents the most comprehensive assessment of genetic diversity and relatedness of malaria infections among ANC users to date.

This study is limited by the number of samples available to sequence from children, particularly when stratifying by site and year, restricting comparisons with ANC users. For intra-host diversity, we

therefore focused on Magude, where most samples from children originated. We did not consider the potential issue of parasite importation from neighboring regions, nor reasons for ANC non-attendance, although we would not expect any potential selection bias[15] to affect the parasite population. Data on previous malaria infection and treatment was not available, and this study was, therefore, limited to single time-point assessment of malaria infection. To confirm the generalizability of this approach for routine surveillance, more studies should be carried out in different epidemiological settings and include larger community sample sizes. Finally, we observed a clear dependence of sequencing coverage on parasite density, which

**Fig. 3 | Population-level *Plasmodium falciparum* genetic diversity among first antenatal care users and children. A** Expected heterozygosity (*He*) and 95% confidence interval for each microhaplotype (*n* = 165) across the population of pregnant women attending their first antenatal care visit (*n* = 289) estimated with MOIRE (R package). Mean $H_E$ across all loci indicated with a dotted line. Darker shade of green reflects higher diversity. **B** Number of distinct alleles observed for each locus (ordered by increasing $H_E$ as in **A**). **C** $H_E$ per locus for each area among ANC users. A single random subsampling without replacement of Magude and Ilha Josina was performed to balance sample size with Manhiça (*n* = 64). **D** Per-locus difference in $H_E$ between Magude and Ilha Josina. Overall difference between the two areas assessed with a linear mixed model with random intercepts and slopes per locus. **E** $H_E$ per locus for 2017 and 2018–2019 among ANC users. Random subsampling of 2017 performed to balance sample size with 2018–2019 (*n* = 123). Loci connected between years by gray lines. **F** Per-locus difference (Δ) in $H_E$

between years. Overall difference between years assessed with a linear mixed model with random intercepts and slopes per locus. **G** $H_E$ per locus for ANC users (pink) and children aged 2–10 years from the community (light blue) in overlapping years. Random subsampling of ANC users performed to balance sample size with children (*n* = 33), matching area of residence. **H** Per-locus difference in $H_E$ between children and ANC users. Overall difference between the two areas assessed with a linear mixed model with random intercepts and slopes per locus. Boxes indicate the 25th and 75th percentiles with the centre line indicating the median, and the whiskers indicate the smallest value within 1.5 times interquartile range below the 25th percentile, and the largest value within 1.5 times the interquartile range above 75th percentile. *P* values in **C**, **E**–**G** are from F-tests. Adjusted for multiple testing using the Benjamin–Hochberg method, a *p* value of <0.0062 indicates statistical significance.

**Table 3 | Prevalence of *Plasmodium falciparum* drug resistance markers among first antenatal care users and children**

| Gene | Haplotype/marker | All | First ANC | Children | |
|---|---|---|---|---|---|
| | | n/N (%) | n/N (%) | n/N (%) | p |
| *pfk13* | | | | | |
| | 446-458-476-493 | 0/343 (0.0%) | 0/260 (0.0%) | 0/83 (0.0%) | 1.00 |
| | 539-543-553-561-574-580 | 0/341 (0.0%) | 0/258 (0.0%) | 0/83 (0.0%) | 1.00 |
| *pfdhps* | | | | | |
| | I431V | 0/374 (0.0%) | 0/284 (0%) | 0/90 (0.0%) | 1.00 |
| | A437G | 351/365 (96.2%) | 266/276 (96.4%) | 85/89 (95.5%) | 0.96 |
| | K540E | 335/355 (94.4%) | 256/269 (95.2%) | 79/86 (91.9%) | 0.37 |
| | A581G | 0/372 (0.0%) | 0/283 (0.0%) | 0/89 (0.0%) | 1.00 |
| | double 437-540 | 285/300 (95.0%) | 220/231 (95.2%) | 65/69 (94.2%) | 0.75 |
| *pfdhfr* | | | | | |
| | N51I | 349/352 (99.1%) | 261/264 (98.9%) | 88/88 (100%) | 0.58 |
| | C59R | 333/337 (98.8%) | 247/251 (98.4%) | 86/86 (100%) | 0.58 |
| | S108N | 350/353 (99.2%) | 265/268 (98.9%) | 85/85 (100%) | 1.00 |
| | triple 51-59-108 | 282/286 (98.6%) | 210/214 (98.1%) | 72/72 (100%) | 0.58 |
| *pfdhps-pfdhfr* | | | | | |
| | quíntuple 51-59-108-437-540 | 233/245 (95.1%) | 179/187 (95.7%) | 54/58 (93.1%) | 0.49 |
| *pfmdr1* | | | | | |
| | N86Y | 4/335 (1.2%) | 3/255 (1.2%) | 1/80 (1.3%) | 1.00 |
| | F184Y | 251/333 (75.4%) | 194/252 (77.0%) | 57/81 (70.1%) | 0.38 |
| | D1246Y | 1/377 (0.3%) | 1/287 (0.3%) | 0/90 (0.0%) | 1.00 |
| *pfcrt* | | | | | |
| | 72-76 CVIET | 4/357 (1.1%) | 1/273 (0.4%) | 3/84 (3.6%) | 0.042 |

*n* number of individuals carrying a mutant allele, *N* total individuals with a valid call. In individuals carrying multiple different genotypes at a given locus, both alleles are considered valid if eMOI>1.1, otherwise only the major allele is included. *p* values from Pearson's chi-square test or Fisher exact test of difference in proportion between ANC and children, depending on sample size.

may be explained by technical limitations. When only few, if any, parasite genomes are present in DNA extracted from a DBS, it will be difficult to amplify the parasite DNA for sequencing. This limitation applies to all genotyping techniques[9], and we reached comparably high sensitivity with the protocol applied here. The relationship between parasite density and intra-host diversity may also be affected by biological processes, such as competitive stress and host immunity[32], and future studies are needed to investigate this. Regardless of underlying causes, parasite density is an important confounding factor to adjust for when studying intra-host diversity.

In conclusion, this study extends the scope of ANC-based sentinel surveillance to include genomic malaria surveillance. We did not observe differences in resistance markers between *P. falciparum* collected from ANC users and children representing the community. When adjusting for parasite density, time, and study area, we also did not see differences in genetic diversity or pairwise relatedness between the two populations. In both ANC users and the community,

we found genetic indicators of a recent reduction in the parasite population in an area targeted for elimination, demonstrating the added value of genomic data for impact evaluation. Multiplexed amplicon sequencing has great potential to support decision-makers with genomic intelligence, and adopting a cost-effective and convenient ANC-based sampling strategy would be a valuable step towards making genomic surveillance more feasible in malaria-endemic areas.

## Methods

### Study design and setting

This genomic surveillance study took place between 2015 and 2019 in three malaria-endemic areas in Maputo province in southern Mozambique (Supplementary Methods). Transmission intensity ranged from low in Manhiça and Magude, to moderate-to-high in Ilha Josina, and it declined in all three areas during the study[17]. Magude district was subject to a package of interventions in 2015–2018

including MDA with dihydroartemisinin- piperaquine and IRS with dichlorodiphenyl-trichloroethane and pirimiphos-methyl, resulting in a 85% reduction of in all-age *P. falciparum* parasite rates[24] (Supplementary Methods p 3).

## Study participants

Samples were collected from pregnant women at ANC clinics and children participating in household surveys. 10,439 pregnant women were recruited when attending their first ANC visit at Manhiça District Hospital, Ilha Josina Health Center, or Magude Health Center between November 2016 and November 2019. For 8910 of the visits, informed consent to participate was obtained, and 8745 visits were included in the study[28]. The main reason for exclusion was not residing in the area. Women donated a finger-prick drop blood onto filter paper (dried blood spot), and HIV status, date, age, gravidity, area of residence, and recent movements were recorded. 6471 samples (74%) were selected for molecular analysis in order to determine *P. falciparum* parasite rates in the three sites with a margin of error lower than or equal to expected parasite rates (5–20%). 483 (7.5%) of these were positive for *P. falciparum* by qPCR. 9381 children aged 2–10 years were sampled for annual age-stratified household surveys in the study area. The surveys were conducted around May every year (following the rainy season) from 2015 to 2019. DBS were obtained together with basic sociodemographic, clinical, and vector-control information[24]. Self-reported gender was evenly represented in the surveys, with 50.3% girls and 49.2% boys (information for remaining 0.5% was not available). 6767 (72.1%) of the DBS were selected for molecular analysis. 366 (5.4%) of the samples were positive by qPCR. This study was a secondary analysis of DBS left from previous analyses. 378 (78.3%) and 180 (49.1%) of samples from ANC users and children, respectively had enough material left for sequencing (at least a third of a DBS).

## Inclusion and ethics

The team of authors combine researchers from Mozambique and non-malaria endemic areas. Mozambican researchers were involved throughout the research process. The research question aims to improve local malaria surveillance and is relevant to the local communities affected by malaria. All study protocols were approved by CISM's and Hospital Clínic of Barcelona's ethics committees, and the Mozambican Ministry of Health National Bioethics Committee. All study participants gave written informed consent, or in the case of minors, written informed assent and consent by a parent/guardian.

## Amplicon sequencing

DNA was extracted from 558 available *P. falciparum*-positive DBS samples (from 378 ANC users and 180 children) using a Tween-Chelex based protocol (Supplementary Methods p 3). A multiplex panel of PCR primers targeting 241 *P. falciparum* amplicons of 150–250 bp was developed (Paragon Genomics Inc, California, USA) (Supplementary Data 1). Amplicons included 165 microhaplotypes informative about genomic diversity and relatedness in southern Africa[25,31], and markers of drug resistance in 15 genes[4], including polymorphisms associated with resistance to artemisinin (*pfk13*), SP (*pfdhfr* and *pfdhps* genes), and amodiaquine (*pfcrt* and *pfmdr1* genes). We followed the Clean-Plex® protocol (Paragon Genomics Inc, USA) with some modifications (Supplementary Methods p 3). Briefly, DNA was amplified for 15 or 20 cycles for multiplexed PCR, depending on parasitemia and ability to amplify, and for 15 cycles for indexing PCR. A randomly selected subset of resulting libraries was assessed by capillary electrophoresis using a TapeStation (Agilent technologies, California, USA). Libraries were pooled accounting for differences in yield due to parasitemia, and the pool was bead-cleaned using CleanMag® Magnetic Beads at 1× ratio to remove primer dimers. Pooled libraries were run on an agarose gel from which the amplicon-sized band was excised, and DNA extracted

using Monarch® DNA Gel Extraction Kit (New England Biolabs Inc., Massachusetts, USA). Library pools were quantified and assessed using a TapeStation and a Qubit fluorometer. The purified libraries were sequenced in either a MiniSeq, or NextSeq instrument (Illumina, San Diego, USA).

## Bioinformatics and data filtering

FASTQ files were run through a Nextflow-based pipeline[33] (version 0.1.5), to infer alleles. Briefly, reads were demultiplexed for each locus using cutadapt[34], and DADA2[35] was used to cluster reads using an error-inference model. Reads were filtered and truncated based on quality and length, also using cutadapt and DADA2. To reduce false positives, homopolymers and tandem repeats were masked. The resulting allele table was subsequently filtered based on read counts and coverage across loci within a sample and across samples. Alleles with fewer reads than the maximum observed reads in any locus for negative controls (14 reads) were removed, along with alleles with <1% within-sample frequency. Samples with a coverage of <50 diversity loci with a read depth of 100 were filtered out. Finally, diversity loci with <100 samples covering them with a read depth of 100 were also removed.

## Definitions

Rainy season was defined as November 1st to April 30th, and the remaining year as dry season[28]. Years were defined based on transmission season, i.e., from November 1st to October 31st. When comparing time periods for ANC users, 2018 and 2019 were combined to balance sample size with 2017, where more cases were sampled due to higher transmission. Only children were sampled in 2015 and 2016, and these years were also combined. Primigravidity was defined as a first pregnancy, while multigravidity was defined as having had at least one previous pregnancy. Population diversity was measured as $H_E$, i.e., the probability that two randomly selected parasites carry distinct alleles at each diversity locus ($n = 165$). It was calculated as:

$$H_E = \left(\frac{n}{n-1}\right)\left(1 - \sum_i p_i^2\right) \quad (1)$$

where $n$ is the population size, and $p_i$ is the frequency of the $i$th allele, with allele frequencies estimated statistically using a Markov chain Monte Carlo (MCMC) algorithm from MOIRE version 3.0.0 (R package)[36,37]. MOIRE source code is available at https://github.com/EPPIcenter/moire. Briefly, MOIRE uses a Bayesian approach to estimate allele frequencies, within-host relatedness, and within-host diversity from polyallelic data subject to experimental error. For the MCMC, we used 10,000 burnin samples followed by 10,000 samples run across 20 parallel tempered chains. We set alpha and beta to 0.1 and 9.9 for false positive rates, and 0.1 and 9.9 for false negative rates, respectively, shape and scale to 0.1 and 10.0 for mean COI, and alpha and beta to 1.0 and 1.0 for relatedness. Intra-host diversity was measured using the following metrics: MOI, eMOI, 1-$F_{ws}$, and proportion of polyclonal infections (eMOI>1.1). Individual MOI and eMOI was also estimated with the MOIRE MCMC algorithm, accounting for allele frequencies across loci. eMOI furthermore takes within-host relatedness ($r$) into account, and can be interpreted as the expected MOI if population diversity was infinite ($H_E = 1$). It is calculated as:

$$eMOI = 1 + (1 - r)(MOI - 1) \quad (2)$$

1-$F_{ws}$ was calculated as the allele heterozygosity of the individual ($H_W$) relative to the population[38]:

$$1 - F_{WS} = \frac{H_W}{H_E} \quad (3)$$

We used 1-$F_{ws}$ in order to have increasing values with increasing diversity, aligned with the other metrics used. $H_W$ is defined as:

$$H_W = 1 - \left( n_i \left( \frac{1}{n_i} \right)^2 \right) \qquad (4)$$

where $n$ is the number of alleles detected at the $i$th locus of a given sample. Individual mean 1-$F_{ws}$ was calculated across all diversity loci. Pairwise infection (inter-host) relatedness was estimated as IBD, i.e., the proportion of the genome shared between parasites through recent ancestry, using Dcifer version 1.2.0 (R package)[39], accounting for the presence of multiple strains in one infection and the probability that regions of the genome are shared by chance. Prevalence of resistance markers were calculated as the number of individuals carrying a mutated allele out of all individuals with a valid genotype for the respective locus. In case of both wildtype and mutant alleles present in one individual, the individual was considered mutant carrier if the infection was polyclonal by eMOI (eMOI > 1.1), otherwise only the major allele (wildtype or mutant) was considered. For genotypes involving multiple amplicons, only samples with a single allele present were included to avoid issues with phasing.

## Statistical analysis
Univariate and multivariate regression analyses were used to estimate intra-host diversity and assess the effect of factors of interest. $P$ values and confidence intervals for eMOI were obtained from zero-truncated Poisson regressions in order to restrict eMOI to positive values. Logistic regression was used for percentage polyclonal and 1-$F_{ws}$. The effect size of continuous time on intra-host diversity was estimated from multivariate regressions with an interaction between time and area. To compare intra-host diversity between ANC users and children, only samples from Magude were included due to low sample sizes for children in Manhiça and Ilha Josina. $H_E$ was compared between populations with Linear Mixed Models (R package nlme version 3.1) fitting locus as a random effect. Simple random subsampling without replacement matching populations by area and year was performed to compare groups of similar sample size. Differences in mean relatedness between areas and populations were assessed with permutation testing (10,000 permutations). Prevalence of resistance markers were compared with Pearson's chi-square test or Fisher's exact test. All statistical tests were two-sided, except the permutation test for mean relatedness. Multiple comparisons were corrected for using the Benjamin-Hochberg procedure with a $q$ value of 0.05, resulting in a final alpha of 0.0062 applied to indicate significance. All analyses were performed using R version 4.3.0.

## Reporting summary
Further information on research design is available in the Nature Portfolio Reporting Summary linked to this article.

## Data availability
Sequencing data and some epidemiological meta data is available at NCBI Sequence Read Archive (SRA) under accession code PRJNA1040019. Sequencing data was aligned to selected regions of the Pf3D7 genome [https://github.com/EPPIcenter/mad4hatter/blob/main/resources/v4/ALL_refseq.fa.] The raw epidemiological data includes personal data and is protected by data privacy laws. This data can be made available for research purposes from the corresponding author, AM, upon reasonable request by email. Requests for data will be reviewed in a three-month timeframe by Manhiça Health Research Center to verify that data sharing is not subject to any intellectual property or confidentiality obligations. If data can be shared, it will be released via a Data Transfer Agreement.

## Code availability
Code for the use of MOIRE R package is available at https://github.com/EPPIcenter/moire, which is licensed under the licensed under GNU General Public License v3.0. R code for data analysis is available upon request. Requests should be addressed to AM (corresponding author).

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

## Acknowledgements

We are grateful to the women, children and their families who participated in the study, the clinical teams at the health centres, and the field and lab teams at CISM and ISGlobal. In addition, we want to acknowledge the developers of MOIRE, Maxwell Murphy, and Dcifer, Inna Gerlovina, who supported us in using their R packages and interpreting the results. Finally, we would also like to express our gratitude to the communities of Magude and Manhiça districts, and the district authorities for their support of the project. This work was supported financially by the Bill and Melinda Gates Foundation (INV-019032, A.M. and OPP1132226, B.G.), the National Institute of Health and National Institute of Allergy and Infectious Diseases (1R01AI123050, A.M./E.M. and K24AI144048, B.G.), the "la Caixa" Foundation (ID 100010434, fellowship code LCF/BQ/DI20/11780016, N.B.), the Departament d'Universitats i Recerca de la Generalitat de Catalunya (AGAUR; grant 2021 SGR 01517, A.M.), Ministerio de Ciencia e Innovación (PID2020-118328RB-I00/AEI/10.13039/501100011033, A.M.), the European Union's Horizon 2020 research and innovation programme (Marie Skłodowska-Curie grant agreement 890477, A.P.). We also acknowledge support from the Spanish Ministry of Science and Innovation and State Research Agency through the "Centro de Excelencia Severo Ochoa 2019–2023" program (CEX2018-000806-S), and support from the Generalitat de Catalunya through the CERCA Program. This research is part of the ISGlobal's Program on the Molecular Mechanisms of Malaria which is partially supported by the Fundación Ramón Areces. CISM is supported by the Government of Mozambique and the Spanish Agency for International Development (AECID). The funders had no role in study design, data collection and analysis, decision to publish, or preparation of the manuscript. The funders of the study had no role in study design, data collection, data analysis, data interpretation, or writing of the report.

## Author contributions

A.M. and E.M. designed the study. S.M., H.Mbeve, N.N., B.C., C.M., P.C, B.Galatas, C.G., H.Munguambe, L.N., W.S., J.M., P.A., A.C., B.Galatas, and F.S. coordinated field work activities and managed samples and data. N.B., C.d.S, and E.N.V. analyzed the samples. B.P. and M.G. processed the raw sequencing files. N.B. analyzed and interpreted the data. A.A.-D., B.Greenhouse., A.P., A.M., G.M. and E.R.-V. contributed to the interpretation of data. N.B. drafted the manuscript. All authors revised the manuscript, and agreed to submit it for publication.

## Competing interests

The authors declare no competing interests.
