## [Peer Review File · Nature Communications]

Genomic malaria surveillance of antenatal care users detects reduced transmission following elimination interventions in MozambiqueReviewers' Comments:

Reviewer #1:

Remarks to the Author:

Brokhattingen et al demonstrate the utility of malaria genomic surveillance on samples obtained from the first ANC visit in Mozambique, providing evidence of the ease of using this population to monitor parasite genetic diversity and drug resistance mutations in the population. They show that the ANC population is representative of the parasite population that is circulating in the community, represented by malaria infections in children.

Overall, the manuscript is well written and clearly highlights the strengths of sampling the ANC population.

Section – Sequencing performance

Line 79 it should be made clear here whether all 241 amplicons and 165 microhaplotypes were obtained for all the 558 samples?

Line 81, when the authors refer to the samples being successfully sequenced does this mean data was obtained for all 241 amplicons and 165 microhaplotypes? In table 1 the 382 reflects the samples that were included and passed the filtering criteria. The inclusion of samples and the filtering criteria are two different things, first the number of samples sequenced and then second the next step, what moves into the bioinformatics pipeline and generates the final data set of microhaplotypes. This should be clearly articulated for the reader to follow in this section.

Line 78-79 grammar error 'were attempted sequenced' this should be changed

Line 79-80 grammar error 'informative about genetic diversity' do the authors mean that provided information on parasite population genetic diversity?

Line 83 grammar "across all samples attempted sequenced" needs to be corrected and the point being made in the sentence relates to table 1 and thus the sentence should move before the sentence that starts "Sequencing performance i.e. total...." to make it easier to follow the sentence. The sentence "Sequencing performance i.e. total...." should move after the sentence in line 85 beginning "Parasite densities were similar.."

Line 89, total reads per sample of 453,541 and median 208 loci covered, line 90 'On average, each locus (n=241) was covered by 1.4 million reads and 462 samples', I cannot pick out these numbers from table 1 or figure 1. They can be highlighted in the figure 1 legend text if appropriate.

Line 95, MOI of 2.4 for ANC attendees does not reflect the values in table 2 that is 1.8? the authors should clarify where the 2.4 value is from.

Section on temporal trends in intra-host genetic diversity

Magude had a decline in eMOI and a comparison was done between ANC and children populations.

IIha Josina had an increasing trend over time in polyclonal infections, Fig 2a should be cited at the end of the sentence. However, why was no further analysis done on IIha Josina?

Line 121 the significant declines in eMOI -36% and -50% for children and ANC attendees respectively, is this from Supplementary table 3? If so, the values do not match or the table can be cited at the end of the sentence.

Lines 131-132, should this be Fig 3c? what does the p-value on the graph represent, if no change was observed between Magude and IIha?

Lines 135-136, unless I have misunderstood the figure, the mean H_e should be figure 3e and 3f rather than 3g and 3h. The authors can check that the right figures are cited next to the right text.

Line 141, mean pairwise IBD of 0.026, I presume is obtained from supplementary table 6? Can this be stated clearly in the text how it is calculated, or in the supplementary Fig 3 legend text or a footnote under supplementary table 6

Line 151, "Prevalence of all markers of antimalarial resistance researched in this study was similar between ANC..." the grammar is incorrect perhaps write 'The prevalence of all markers of antimalarial resistance assessed in this study were similar between ANC..'

Line 196, 'we found the highest population diversity in the highest transmission setting' should this read complexity of infections as IIha had more polyclonal infections?

Line 243, 'the relationship between density and intra-host...' add the word parasite before density for clarity.

In the methods section, under study participants, were the DBS taken from everyone or only malaria positive individuals? It is not clear how from 8745 for ANC the 378 samples were selected for genotyping? Also, moving from 3933 down to 180 DBS needs clarification.

In the section on bioinformatics and data filtering, the authors should be clear of what the filtering steps are and are these additional steps after the Q score filters? Line 313 states finally, however the next sentence continues to describe steps that appear to be part of the filtering process it should state it is part of the criteria for filtering to include or exclude samples for the next sets of analysis e.g. MOI, drug resistance markers etc

Reviewer #2:

Remarks to the Author:

The manuscript is appropriate for acceptance with edits. This manuscript will contribute to the field of malaria surveillance in high/low transmission settings using first antenatal care (ANC) visits as a sentinel cohort, its importance and approach to describe it. This study shows that using the ANC cohort makes possible to follow the transmission and prevalence of disease in a cost-effective and convenient way and shows comparative analysis with the children cohort (age 2-10 years) from overlapping regions and time. This is one of the first studies in the region of sub-Saharan Africa to use as source the ANC cohort with larger sample sizes. This study makes available new genotype data for *P. falciparum* population in southern Mozambique. Most of the conclusions of the study are supported by/interpreted based on the results, but can be expanded.

Minor points/questions/suggestions:

1. General,

- a. A separate table can be provided to show the number of samples used in different comparisons.
- b. A map of southern Mozambique can be provided to show the location of the three studied regions. It can help in interpreting the cross-transmission and distance between locations.
- c. Is information on previous malaria infection(s) and treatment available for both cohorts? This can help in understanding the heterozygosity.

2. In Results section,

- a. In line 79-80, references for 165 microhaplotypes and drug resistance markers should be mentioned.
- b. In line 83, number of total loci should be mentioned.
- c. In line 83-84, the statement is not justified based on the result Figure 1C.
- d. In line 132-133, to compare the HE between populations, the random sampling was done only once or multiple random samplings were done to get significance of comparisons?
- e. In line 140, each strain in each polyclonal infection was considered independent. How were the strains defined or deconvoluted from polyclonal infection genotypic data?
- f. In line 221-224, is better performance of accuracy and sensitivity of microhaplotypes than SNP based approaches valid for all transmission setting scenarios? Are the results of this study compared to any available WGS data from Mozambique (2015-2019)?
- g. In line 224-226, SNP/WGS based approaches can utilize polyclonal infections based on deconvolution and other available methods to either select the major allele or alleles of independent strains.
- h. In Supplementary table 6, what was permuted?

Reviewer #3:

Remarks to the Author:

Nanna Brokhattingen and colleagues have explored whether amplicon sequencing of *P. falciparum* infections collected during antenatal care (ANC) visits is a representative and informative approach for malaria epidemiological surveillance. The two main conclusions are: (i) infections from ANC visits are equivalent, in terms of genetic diversity and drug resistance prevalence, to infections from cross-sectional surveys of children; (ii) genetic diversity metrics reflect local changes in transmission driven by control interventions.

Overall, I really enjoyed reading the manuscript and felt it systematically explored an interesting and valuable question. The sequencing results are impressive given the asymptomatic and low parasitemia samples, and it is clear thought has been put into study design. However, I do have some major comments I would like to see addressed and several minor comments.

Major Comments

1. One of the main conclusions from the paper is that there is a relationship between genetic data and epidemiological data in southern Mozambique. The epidemiological data is half of this argument and yet it is only described in the text – it is not featured in any figure or table, main or supplementary. Incidentally, the existing text descriptions are somewhat confusing as they cite different studies, e.g. in the discussion Manhica is stated as having an 8- to 3% decline in P.f. positivity over the study period, and the supplementary methods state the decline is from 5% to 0%. Other manuscripts exploring relationships between genetics and epidemiology have made direct comparisons and included the epidemiological data in the main display items (e.g. <https://www.nature.com/articles/s41467-023-43087-4>). A similar effort should be made here.
2. The authors mention that transmission declined in all three study sites during the study period, yet intra-host diversity only declines in Magude, where interventions were deployed. This is an interesting and critical observation, as it complicates the conclusion that genetics detected reduced transmission. It might suggest that the genetic diversity statistics were more directly impacted by the interventions deployed, with the transmission decline being correlation. I think addressing the first comment will help, but the authors should grapple with this observation more directly. The current sentence starting on Line 216, even if true, doesn't attempt to address why genetic signals of decline were not observed in the other two sites.
3. I am convinced that the population genetic diversity and drug-resistance prevalence is the same between the ANC visits / children. However, I am less convinced that the intra-host diversity statistics are equivalent. For e.g., in Table 2 the polyclonal fraction in Manhica 47% in ANC vs. 70% in children. I can see that with the current sample sizes, estimated 95% CIs overlap; but to me it is not clear how this will shake out when more samples are collected. There also isn't a temporal trend in polyclonal fraction for children (but one is claimed in Line 123?), which may be because the COI distribution is shifted upwards. Children may have different exposure &/or immunity which may impact absolute number of clones. I don't think the data support a conclusion as strong as the one in the sentence starting on Line 250.
4. Many of the findings in the manuscript rely on genetic diversity measures computed from the software package MOIRE. The citation is to a github repository, which has a link to a recent preprint; it seems it has not been through peer review. Of course, not every software package that is used in a manuscript needs to have been peer reviewed. Here, however, it seems the software package implements an entirely new statistical model for joint inference of COI and intra-host relatedness from multiallelic amplicon sequencing data, as well as defines a novel measure of intra-host diversity (eMOI). As a reviewer, this feels quite awkward/unfair: either I proceed on trust that the model, inferences, and new statistic that underlie your results are all sound; or I have to try and additionally review the MOIRE preprint. At the same time, there exist other published software for intra-host diversity analysis (e.g. THEREALMcCOIL). If the aim is to publish this work before MOIRE is peer-reviewed, could you not support some of your critical findings with established software and statistics? For example, a key finding is the decline in intra-host diversity in Magude in both ANC and children. If you compute COI with THEREALMcCOIL or similar, can you show that this result still holds?
5. The formula for F_{ws} is incorrect – it seems instead you have another formula for within-host

heterozygosity? But this formula too I struggle to understand.

Minor Points

- Line 77 and elsewhere, the phrase "Attempted sequenced" should just be "sequenced"; or maybe "selected for sequencing".
- Line 97, is this the same Fws statistic defined by Magnus Manske et al. (<https://www.nature.com/articles/nature11174>)? If so, I think you should cite that manuscript. Also, in personal experience, I've never heard this statistic described as "1 - Wright's inbreeding coefficient", rather just "Fws", the latter also being more concise
- In Figure 1, please indicate in the figure caption what quantiles are demarcated by the box and whiskers in the boxplots
- In the caption of Figure 1B, could you please precisely describe what is meant by "covered"? E.g. is a locus considered covered when it has a minimum of 1X coverage across all its bases?
- The information in Figure 1D/E is important but the presentation could be improved. For example, in Figure 1D, the x-axis position of the amplicons has no indicated significance; rather just the label "Diversity". If there is a significance, please indicate that. If not, the plot would be more informative if, for example, you ordered the amplicons by their median abundance, or the chromosome they are on, or both; anything that you think would help readers better appreciate the contents of your panel and/or the variation in sequencing performance across amplicons. Similarly, for Figure 1E, you could break "Resistance" by loci.
- Figure 2 caption, states "eMOI in Magude in 0-truncated Poisson regression adjusted for parasitemia", but p-values are shown for all three regions.
- Figure 2 caption, "estimated with Poisson and logistic regressions similar to A-B", should be A-C.
- Throughout Population genetic diversity, the text and the figure panel labels do not match. For example, Fig. 3c,d is indicated when describing temporal trends; whereas this should be Fig. 3e.
- Figure 3 caption and Line 359, when you randomly subsample, how many replicates do you perform? Also I would like to confirm this is random sampling without replacement.
- Line 337, "Multiple Chain" should be "Markov Chain"
- Consider including the mathematical definition of eMOI in the Definitions section of the methods; personally, I found it much easier to understand than the current verbal description.
- If eMOI takes continuous values, could you please comment on why you've chosen to do Poisson regression

We would like to thank the editors and reviewers for their time and constructive comments. Below, please find our responses in italics to each reviewer comment.

Reviewer #1:

Brokhattingen et al demonstrate the utility of malaria genomic surveillance on samples obtained from the first ANC visit in Mozambique, providing evidence of the ease of using this population to monitor parasite genetic diversity and drug resistance mutations in the population. They show that the ANC population is representative of the parasite population that is circulating in the community, represented by malaria infections in children.

Overall, the manuscript is well written and clearly highlights the strengths of sampling the ANC population.

Line 79 it should be made clear here whether all 241 amplicons and 165 microhaplotypes were obtained for all the 558 samples?

RESPONSE: *We have now included this information in the sentence “On average, quality reads were obtained for 212 and 170 loci per sample for ANC users and children, respectively (Table 1).”*

Line 81, when the authors refer to the samples being successfully sequenced does this mean data was obtained for all 241 amplicons and 165 microhaplotypes? In table 1 the 382 reflects the samples that were included and passed the filtering criteria. The inclusion of samples and the filtering criteria are two different things, first the number of samples sequenced and then second the next step, what moves into the bioinformatics pipeline and generates the final data set of microhaplotypes. This should be clearly articulated for the reader to follow in this section.

RESPONSE: *We agree and have changed the categories in the table to “Passed filtering” and “Filtered out”. Similarly in the text, we have changed the category names.*

Line 78-79 grammar error ‘were attempted sequenced’ this should be changed

RESPONSE: *Agree. “Attempted” has been deleted so that it simply states the number of samples that were sequenced, and the number of samples that passed filtering.*

Line 79-80 grammar error ‘informative about genetic diversity’ do the authors mean that provided information on parasite population genetic diversity?

RESPONSE: *Yes, exactly. We have changed the wording to “241 amplicons in the P. falciparum genome were targeted, including 165 microhaplotypes that had been identified to provide information about genetic diversity in the parasite population”.*

Line 83 grammar “across all samples attempted sequenced” needs to be corrected and the point being made in the sentence relates to table 1 and thus the sentence should move before

the sentence that starts “Sequencing performance i.e. total....” to make it easier to follow the sentence. The sentence “Sequencing performance i.e. total....” should move after the sentence in line 85 beginning “Parasite densities were similar...”

RESPONSE: *We apologize that this was not clear, and have moved the sentences around as suggested.*

Line 89, total reads per sample of 453,541 and median 208 loci covered, line 90 ‘On average, each locus (n=241) was covered by 1.4 million reads and 462 samples’, I cannot pick out these numbers from table 1 or figure 1. They can be highlighted in the figure 1 legend text if appropriate.

RESPONSE: *Thanks for pointing out the inconsistency. We have changed the sentence to report these numbers by population in accordance with Table 1: “Sequencing coverage was high across samples that passed filtering (n=328), with a geometric mean total reads per sample of 319,503 and 484,245, and a median 223 and 213 out of 241 loci covered per sample, for ANC users and children, respectively (Table 1)”. We have also added a sentence in the Figure 1 legend about the number of reads per locus: “On average, each locus was covered by 1.4 million reads.”.*

Line 95, MOI of 2.4 for ANC attendees does not reflect the values in table 2 that is 1.8? the authors should clarify where the 2.4 value is from.

RESPONSE: *The 2.4 refers to MOI, which is not adjusted for relatedness, while 1.8 refers to effective MOI (eMOI), which has been adjusted for intra-host relatedness. We have included a supplementary table with regular MOI, and referred to this, and clarified the use of unadjusted and adjusted MOI in the results: “On average, ANC attendees had a multiplicity of infection (MOI; unadjusted for relatedness) of 2.37, i.e., carried 2.37 genetically different *P. falciparum* parasite clones (Supplementary Table 1). $1-F_{ws}$ was 0.39, while effective multiplicity of infection (eMOI), which incorporates intra-host relatedness between clones, was lower than MOI at 1.8, both indicative of inbreeding (Supplementary Fig. 1).”*

Magude had a decline in eMOI and a comparison was done between ANC and children populations. Ilha Josina had an increasing trend over time in polyclonal infections, Fig 2a should be cited at the end of the sentence. However, why was no further analysis done on Ilha Josina?

RESPONSE: *Unfortunately, the sample size for children in Ilha Josina was too small (N=4, after filtering) to do further comparisons between ANC users and children in this area. We would like to point out that there was no change over time in multiplicity of infection in Ilha Josina to support an increasing trend in polyclonality in the area, which also was not significant (p=0.33). Nonetheless, it would have been very interesting to compare ANC users in children in the other areas. We hope this study inspires future comparisons between ANC users and the community across different epidemiological settings.*

Line 121 the significant declines in eMOI -36% and -50% for children and ANC attendees respectively, is this from Supplementary table 3? If so, the values do not match or the table can be cited at the end of the sentence.

RESPONSE: Thanks for pointing this out. The numbers in the text should be -37% and -51%, according to Table 3. This has now been corrected.

Lines 131-132, should this be Fig 3c? what does the p-value on the graph represent, if no change was observed between Magude and Ilha?

RESPONSE: This should be Fig 3g,h. Apologies for the mistake, we have now corrected it. The p-value on Fig 3c ($p=0.0004$) refers to the difference in expected heterozygosity (HE) between Magude and Ilha Josina in a linear mixed model including locus as a random intercept, so there was indeed a difference between these two areas. The p-value in Fig 3g ($p=0.95$) refers to the difference between HE among ANC users and children, using a similar model.

Lines 135-136, unless I have misunderstood the figure, the mean He should be figure 3e and 3f rather than 3g and 3h. The authors can check that the right figures are cited next to the right text.

RESPONSE: Apologies for this mixing up, we have now corrected it.

Line 141, mean pairwise IBD of 0.026, I presume is obtained from supplementary table 6? Can this be stated clearly in the text how it is calculated, or in the supplementary Fig 3 legend text or a footnote under supplementary table 6

RESPONSE: We have elaborated on how IBD is estimated in the Supplementary Table 6 legend (“Identity by descent (IBD), i.e., pairwise relatedness between infections, was estimated with Dcifer (R package).⁴ All infections, including unphased polyclonal ones, are included. To test for differences in mean IBD between and within areas, we performed permutation of area labels (10,000 permutations) and compared mean IBD with permutation distributions. P-values are from permutation tests.”) and the results text (“Genetic relatedness between pairs of *P. falciparum* infections, including unphased polyclonal infections, was estimated with an identity by descent (IBD)-based approach. To compare relatedness between areas and populations, we performed permutation of labels and compared mean IBD in each area or population with permutation distributions.”), and referenced an article describing the R tool, Dcifer, that we used to estimate it.

Line 151, “Prevalence of all markers of antimalarial resistance researched in this study was similar between ANC...” the grammar is incorrect perhaps write ‘The prevalence of all markers of antimalarial resistance assessed in this study were similar between ANC..’

RESPONSE: We have corrected the sentence.

Line 196, ‘we found the highest population diversity in the highest transmission setting’ should this read complexity of infections as Ilha had more polyclonal infections?

RESPONSE: We were referring to expected heterozygosity, and have now included that in a parenthesis for clarity.

Line 243, ‘the relationship between density and intra-host...’ add the word parasite before density for clarity.

RESPONSE: We have now included “parasite” to avoid any doubt.

In the **methods** section, under study participants, were the DBS taken from everyone or only malaria positive individuals? It is not clear how from 8745 for ANC the 378 samples were selected for genotyping? Also, moving from 3933 down to 180 DBS needs clarification.

RESPONSE: *Apologies for not explaining the numbers more detailed. We have now elaborated on how samples were selected for molecular analysis in the Methods section - Study participants paragraph.*

In the section on **bioinformatics** and data filtering, the authors should be clear of what the filtering steps are and are these additional steps after the Q score filters? Line 313 states finally, however the next sentence continues to describe steps that appear to be part of the filtering process it should state it is part of the criteria for filtering to include or exclude samples for the next sets of analysis e.g. MOI, drug resistance markers etc

RESPONSE: *Thanks for pointing it out. There were two separate filtering processes, one to infer alleles from FASTQ files, and a subsequent one for the resulting alleles. We have tried to provide more separated explanations to highlight this: "FASTQ files were run through a Nextflow-based pipeline (version 0.1.5), to infer alleles [...] The resulting allele table was subsequently filtered based on read counts and coverage across loci within a sample and across samples. [...]"*

Reviewer #2:

The manuscript is appropriate for acceptance with edits. This manuscript will contribute to the field of malaria surveillance in high/low transmission settings using first antenatal care (ANC) visits as a sentinel cohort, its importance and approach to describe it. This study shows that using the ANC cohort makes possible to follow the transmission and prevalence of disease in a cost-effective and convenient way and shows comparative analysis with the children cohort (age 2-10 years) from overlapping regions and time. This is one of the first studies in the region of sub-Saharan Africa to use as source the ANC cohort with larger sample sizes. This study makes available new genotype data for *P. falciparum* population in southern Mozambique. Most of the conclusions of the study are supported by/interpreted based on the results, but can be expanded.

A separate table can be provided to show the number of samples used in different comparisons.

RESPONSE: *We appreciate the suggestion. Where missing, we have now added the number of samples used in different comparisons in figures and tables, or their legends, including supplementary material. We hope that the numbers are sufficiently clear now.*

A map of southern Mozambique can be provided to show the location of the three studied regions. It can help in interpreting the cross-transmission and distance between locations.

RESPONSE: *Thanks for the suggestion. We have now included a map of southern Mozambique in Supplementary methods indicating the three study areas and the distance between them.*

Is information on previous malaria infection(s) and treatment available for both cohorts? This can help in understanding the heterozygosity.

RESPONSE: This information is unfortunately not available. We only have information about current infections. We have added this to the Discussion paragraph about limitations of the study “Data on previous malaria infection and treatment was not available, and this study was, therefore, limited to single time-point assessment of infection.”.

In line 79-80, references for 165 microhaplotypes and drug resistance markers should be mentioned.

RESPONSE: We have now included the relevant reference.

In line 83, number of total loci should be mentioned.

RESPONSE: We agree that this was not clearly explained, and have rephrased the sentence about the number of loci to “We used a panel targeting 241 amplicons in the *P. falciparum* genome. The amplicons included markers of drug resistance in 15 genes and 165 microhaplotypes that had been selected to provide information about genetic diversity in the parasite population.” and hope that the number of loci is clearer now. Thanks for bringing it to our attention.

In line 83-84, the statement is not justified based on the result Figure 1C.

RESPONSE: The statement about parasite densities in the two populations is justified based on the results in Table 1. We have moved the sentences around and referred to the table immediately before this sentence to make it clearer (“On average, quality reads were obtained 212 and 170 loci per sample for ANC users and children, respectively (Table 1). 68.5% (382/558) of the samples passed the filtering criteria. A lower proportion of samples from children passed filtering (51.7%) compared to ANC users (76.5%).”). Thanks for pointing out that it was not clear.

In line 132-133, to compare the HE between populations, the random sampling was done only once or multiple random samplings were done to get significance of comparisons?

RESPONSE: *The random sampling refers to a single random draw of the population of larger size to balance it with the smaller population. This was done without replicates because of the computational expense of running MOIRE to obtain HE for each population. We have elaborated the sentence describing this in the Results section to make it clearer: "In order to compare HE between time windows, areas, and population groups, the larger populations were randomly subsampled within areas and/or years once without replacement to match the smaller population in size.". We have also specified this in the legends of Fig.3 and Supplementary Tables 7 and 8, and in Methods - Statistical analysis. Thanks for pointing out the missing information.*

In line 140, each strain in each polyclonal infection was considered independent. How were the strains defined or deconvoluted from polyclonal infection genotypic data?

RESPONSE: *We realize that this was not stated clearly enough and apologize for the confusion. Each strain in each polyclonal infection was not considered independent. Rather, the method used (Dcifer) provides a single estimate for the overall relatedness between two infections, without phasing/deconvoluting polyclonal infections. We have rephrased the sentence to "Genetic relatedness between pairs of *P. falciparum* infections, including unphased polyclonal infections, was estimated using an identity-by-descent (IBD)-based approach." to highlight that the polyclonal infections are unphased.*

In line 221-224, is better performance of accuracy and sensitivity of microhaplotypes than SNP based approaches valid for all transmission setting scenarios? Are the results of this study compared to any available WGS data from Mozambique (2015-2019)?

RESPONSE: *We have now elaborated on the statement on accuracy and sensitivity of microhaplotypes, and referenced two previous studies that assessed different transmission scenarios in Mozambique and compared amplicon sequencing to WGS, respectively (in bold): "Strengths of this study include the rich data obtained from deep amplicon sequencing, with sensitivity to achieve good coverage for samples with down to 10 parasites/ μ L. **A previous study comparing amplicon sequencing to whole genome sequencing showed that amplicon sequencing provides higher coverage, thereby allowing for more sensitive detection of minority strains, even for infections with high parasite densities.** Compared to single nucleotide polymorphism (SNP)-based methods, microhaplotypes allow for higher resolution and consequently more accurate estimates of diversity and relatedness, while being more convenient than microsatellites. **Microhaplotypes have previously been used in a nation-wide study from Mozambique, where they proved able to distinguish parasites from the northern and southern parts of the country.** Furthermore, although possible to distinguish major and minor alleles in polyclonal infections using SNPs, highly diverse microhaplotypes allow a more accurately assessment and better utilization of the full allele diversity in polyclonal samples, which was half of the samples in this study. Another strength of this study is the large ANC sample size, collected prospectively across three years in three different transmission scenarios. To the best of our knowledge, this study represents the most*

comprehensive assessment of genetic diversity and relatedness of malaria infections among ANC users to date.”

In line 224-226, SNP/WGS based approaches can utilize polyclonal infections based on deconvolution and other available methods to either select the major allele or alleles of independent strains.

RESPONSE: *We agree, and have tried to better capture this point with the sentence “Furthermore, although possible to distinguish major and minor alleles in polyclonal infections using SNPs, highly diverse microhaplotypes allow a more accurate assessment and better utilization of the full allele diversity in polyclonal samples.”*

In Supplementary table 6, what was permuted?

RESPONSE: *We permuted area labels. We have now included a description and elaborated on the permutation procedure in the legends of Supplementary Table 9 and 10, in the Results section - Relatedness paragraph, and in the Methods - Statistical analysis paragraph, respectively. Thanks for pointing out the missing information.*

Reviewer #3:

Nanna Brokhattingen and colleagues have explored whether amplicon sequencing of *P. falciparum* infections collected during antenatal care (ANC) visits is a representative and informative approach for malaria epidemiological surveillance. The two main conclusions are: (i) infections from ANC visits are equivalent, in terms of genetic diversity and drug resistance prevalence, to infections from cross-sectional surveys of children; (ii) genetic diversity metrics reflect local changes in transmission driven by control interventions.

Overall, I really enjoyed reading the manuscript and felt it systematically explored an interesting and valuable question. The sequencing results are impressive given the asymptomatic and low parasitemia samples, and it is clear thought has been put into study design. However, I do have some major comments I would like to see addressed and several minor comments.

One of the main conclusions from the paper is that there is a relationship between genetic data and epidemiological data in southern Mozambique. The epidemiological data is half of this argument and yet it is only described in the text – it is not featured in any figure or table, main or supplementary. Incidentally, the existing text descriptions are somewhat confusing as they cite different studies, e.g. in the discussion Manhiça is stated as having an 8- to 3% decline in P.f. positivity over the study period, and the supplementary methods state the decline is from 5% to 0%. Other manuscripts exploring relationships between genetics and epidemiology have made direct comparisons and included the epidemiological data in the main display items (e.g. <https://www.nature.com/articles/s41467-023-43087-4>). A similar effort should be made here.

RESPONSE: We thank the reviewer for pointing out that we did not put enough emphasis on the epidemiological part of our argument. We have now included *P. falciparum* parasite rates using qPCR among pregnant women and children in Fig.2 to make them more explicit.

Furthermore, we have included a direct comparison between *P. falciparum* parasite rates using qPCR and intra-host diversity in Supplementary Fig. 5 and 6, and Supplementary Tables 5 and 6, respectively, along with a paragraph describing the findings in the Results section: “To assess the potential of using intra-host genetic diversity as a proxy for transmission intensity, we compared mean eMOI, proportion polyclonal, and mean $1-F_{WS}$, with another proxy for transmission intensity, *P. falciparum* parasite rates using quantitative polymerase chain reaction (qPCR) (Fig.2, Supplementary Fig. 5,6, Supplementary Table 5,6). *P. falciparum* parasite rates in ANC users declined in all three areas during the study, starting from a higher rate in Ilha Josina compared to Magude and Manhiça (previously reported in¹⁷). In Magude, eMOI in both ANC users and children showed positive Pearson’s correlation coefficients (PCC) close to 1 (>0.85), although not statistically significant. Furthermore, for ANC users in Magude, both $1-F_{WS}$ and proportion of infections that were polyclonal showed $PCC>0.65$. In the other two areas,

PCC was negative, but small and not statistically significant, for all three metrics of intra-host genetic diversity.”

Comparison between genetic metrics and parasite rates in first antenatal care users (Supplementary Fig. 5):

P. falciparum parasite rates (PR) by quantitative polymerase chain reaction (qPCR) is compared with A. effective multiplicity of infection (eMOI), B. proportion of infections that are polyclonal, and C. 1-Fws in pregnant women at their first antenatal care visit. Each dot represents estimates for one year with 95% confidence intervals. N=120 in Magude, n=64 in Manhiça, n=105 in Ilha Josina.

Comparison between genetic metrics and parasite rates in children in Magude (Supplementary Fig. 6):

P. falciparum parasite rates (PR) by quantitative polymerase chain reaction (qPCR) in children aged 2-10 years sampled in household surveys are compared with A. effective multiplicity of infection (eMOI), B. proportion of infections that are polyclonal, and C. 1-Fws in pregnant women at their first antenatal care visit. Each dot represents estimates for one year with 95% confidence intervals. N=47.

Additionally, we have discussed the findings further in Discussion: “Comparing *P. falciparum* parasite rates with metrics of intra-host genetic diversity, we only observed positive correlations in Magude. This indicates that other factors might affect the relationship between parasite rates

and genetic diversity, such as the temporal scales at which malaria transmission and genetic diversity change. Furthermore, the sites are likely to differ in host immunity due to previous exposure, which in turn might affect the chance of measuring multiple clones in one individual. Similarly, health care seeking behavior and the use of antimalarials might differ between the sites. It cannot be discarded that diversity statistics may be more directly impacted by the interventions deployed in Magude, with the transmission decline being a correlation. Genetic diversity on its own might, therefore, not always be a suitable proxy for local transmission intensity, and stratification based on genetic metrics should be carefully validated against other epidemiological data, including assessing the potential role of importation and underlying reasons for changes to transmission intensity.”

The authors mention that transmission declined in all three study sites during the study period, yet intra-host diversity only declines in Magude, where interventions were deployed. This is an interesting and critical observation, as it complicates the conclusion that genetics detected reduced transmission. It might suggest that the genetic diversity statistics were more directly impacted by the interventions deployed, with the transmission decline being correlation. I think addressing the first comment will help, but the authors should grapple with this observation more directly. The current sentence starting on Line 216, even if true, doesn't attempt to address why genetic signals of decline were not observed in the other two sites.

RESPONSE: *In line with our response to the previous comment, several factors may affect the relationship between parasite rates (as another proxy for transmission intensity) and intra-host genetic diversity, and might explain why the correlation was observed in Magude but not in other sites. We agree with the reviewer that genetic diversity statistics may be more directly impacted by the interventions deployed, with the transmission decline being correlation, and have tried to capture that with the paragraph above.*

I am convinced that the population genetic diversity and drug-resistance prevalence is the same between the ANC visits / children. However, I am less convinced that the intra-host diversity statistics are equivalent. For e.g., in Table 2 the polyclonal fraction in Manhica 47% in ANC vs. 70% in children. I can see that with the current sample sizes, estimated 95% CIs overlap; but to me it is not clear how this will shake out when more samples are collected. There also isn't a temporal trend in polyclonal fraction for children (but one is claimed in Line 123?), which may be because the COI distribution is shifted upwards. Children may have different exposure &/or immunity which may impact absolute number of clones. I don't think the data support a conclusion as strong as the one in the sentence starting on Line 250.

RESPONSE: *Crude comparisons between the average polyclonal fraction of the two populations should be done with caution. Parasite density has a very significant effect on the number of clones detected, so any comparison should be adjusted for this. Secondly, the timing of sampling the two populations is different, i.e. the children are sampled immediately following the rainy season (whereas ANC users were sampled continuously), and sampling started earlier for children than for ANC users (i.e. when transmission was higher). So it makes sense that diversity in children is higher when comparing the crude fraction of polyclonal. In the multiple regression we adjust for parasite density, time and place, and here we do not observe any difference between populations. We have moderated the conclusion to “We did not observe*

differences in resistance markers between P. falciparum collected from ANC users and children representing the community. When adjusting for parasite density, time, and study site, we also did not see differences in genetic diversity or pairwise relatedness between the two populations.”. The claim in line 123 is wrong, we apologize for that, and thank the reviewer for pointing it out. For children, eMOI declines, while only 1-Fws shows a declining trend, and there is no evidence of a temporal trend in fraction polyclonal (only for ANC users). We have removed the statement on polyclonality from the sentence.

Many of the findings in the manuscript rely on genetic diversity measures computed from the software package MOIRE. The citation is to a github repository, which has a link to a recent preprint; it seems it has not been through peer review. Of course, not every software package that is used in a manuscript needs to have been peer reviewed. Here, however, it seems the software package implements an entirely new statistical model for joint inference of COI and intra-host relatedness from multiallelic amplicon sequencing data, as well as defines a novel measure of intra-host diversity (eMOI). As a reviewer, this feels quite awkward/unfair: either I proceed on trust that the model, inferences, and new statistic that underlie your results are all sound; or I have to try and additionally review the MOIRE preprint. At the same time, there exist other published software for intra-host diversity analysis (e.g. THEREALMcCOIL). If the aim is to publish this work before MOIRE is peer-reviewed, could you not support some of your critical findings with established software and statistics? For example, a key finding is the decline in intra-host diversity in Magude in both ANC and children. If you compute COI with THEREALMcCOIL or similar, can you show that this result still holds?

RESPONSE: *We sincerely apologize for this situation. We chose to use MOIRE even before it has been peer-reviewed because we have not been able to find any other software appropriate for our microhaplotype data. THEREALMcCOIL was developed for biallelic SNP-data and is not able to deal with polyallelic data (>2 alleles per locus). Worth mentioning, MOIRE is developed by the same team that developed THEREALMcCOIL, exactly to fill out this gap. Polyallelic loci are also much more informative about MOI and can therefore provide more accurate results than SNPs, which we want to utilize in our analysis (as demonstrated e.g. in Tessema et al. 2020, Fig.4 <https://pubmed.ncbi.nlm.nih.gov/32840625/>). We realize that using a not yet established metric like eMOI (although we are convinced that it comes closer to the true MOI because of its ability to distinguish between related and unrelated clones within an infection) raises doubt about the robustness of our findings. Therefore, we have included MOI estimated using the “naive offset” method, i.e., taking the second highest number of observed alleles in a sample, and “regular” MOI estimated with MOIRE without correcting for intra-host relatedness.*

First ANC users	Naive offset MOI (95% CI)	p	p*	MOI (95% CI)	p	p*	% polyclonal (95% CI)	p	p*
All (n=289)	2.37 (2.20;2.55)			2.37 (2.19;2.55)			60.2 (54.5;65.7)		
Year		0.46	0.16		0.48	0.35		0.54	0.17
2017 (n=166)	2.43 (2.20;2.67)			2.42 (2.19;2.67)			58.4 (50.5;65.9)		
2018-2019 (n=123)	2.29 (1.78;2.94)			2.29 (1.78;2.94)			62.6 (53.4;71.0)		
Change per year	-0.048 (-0.15;0.048)	0.33	0.039	-0.057 (-0.15;0.039)	0.25	0.077	NA	NA	0.11
Area		0.13	0.053		0.20	0.13		0.71	0.21
Magude (n=120)	2.58 (2.31;2.88)			2.56 (2.28;2.86)			62.5 (53.2;71.0)		
Manhiça (n=64)	2.16 (1.57;2.93)			2.20 (1.61;2.99)			56.2 (43.3;68.4)		
Ilha Josina (n=105)	2.26 (1.70;2.98)			2.25 (1.69;2.97)			60.0 (50.0;69.3)		
Parasite density		0.041	0.039		<0.0001	<0.0001		<0.0001	<0.0001
<100 p/μL (n=103)	2.11 (1.84;2.49)			1.74 (1.50;2.01)			42.7 (33.1;52.8)		
100-<1000 p/μL (n=106)	2.39 (1.74;3.26)			2.55 (1.82;3.55)			76.4 (67.0;83.9)		
>=1000 p/μL (n=80)	2.69 (1.94;3.70)			2.94 (2.08;4.12)			61.3 (49.7;71.7)		
HIV		0.38	0.47		0.21	0.35		0.77	0.80
Negative (n=220)	2.33 (2.13;2.53)			2.30 (2.11;2.51)			61.8 (49.1;73.0)		
Positive (n=69)	2.51 (1.93;3.25)			2.57 (1.98;3.32)			59.7 (52.9;66.2)		
Gravidity		0.36	0.64		0.043	0.44		0.11	0.79
Primi (n=117)	2.47 (2.20;2.77)			2.59 (2.31;2.89)			65.8 (56.4;74.2)		
Multi (n=172)	2.30 (1.76;3.00)			2.22 (1.70;2.88)			56.4 (48.6;63.9)		
Season		0.98	0.83		0.57	0.93		0.71	0.45
Dry (n=182)	2.37 (2.15;2.60)			2.41 (2.19;2.64)			59.3 (51.8;66.65)		
Rainy (n=107)	2.37 (1.84;3.04)			2.30 (1.79;2.94)			61.7 (51.7;70.8)		
Children 2-10 years									
All (n=93)	2.70 (2.38;3.05)			2.86 (2.53;3.22)			68.8 (58.8;77.3)		
Year		0.066	0.46		0.012	0.057		0.92	0.45
2015-2016 (n=60)	2.98 (2.57;3.44)			3.23 (2.80;3.71)			70.0 (56.6;80.8)		
2017 (n=4)	2.50 (1.06;5.16)			2.50 (1.07;5.13)			75.0 (21.9;98.7)		
2018-2019 (n=29)	2.14 (1.37;3.27)			2.14 (1.38;3.24)			65.5 (45.7;81.4)		
Change per year	-0.11 (-0.21;-0.015)	0.023	0.38	-0.16 (-0.26;-0.066)	0.0008	0.027	NA	NA	0.27
Area		0.21	0.48		0.092	0.18		0.65	0.15
Magude (n=47)	2.64 (2.20;3.13)			2.91 (2.45;3.43)			68.1 (52.7;80.5)		
Manhiça (n=42)	2.88 (1.87;4.39)			2.95 (1.95;4.43)			73.8 (57.7;85.6)		
I. Josina (n=4)	1.50 (0.49;3.70)			1.25 (0.37;3.23)			25.0 (1.32;78.1)		
Parasite density		0.21	0.19		0.0011	0.0008		0.57	0.68
<100 p/μL (n=33)	2.36 (1.88;2.93)			2.27 (1.80;2.83)			66.7 (48.1;81.4)		
100-<1000 p/μL (n=28)	2.86 (1.66;4.84)			3.00 (1.74;5.10)			78.6 (58.5;91.0)		
>=1000 p/μL (n=20)	3.15 (1.79;5.43)			4.10 (2.37;6.98)			75.0 (50.6;90.4)		

We have analyzed the temporal trends using naive offset MOI and MOI, and find very similar trends to eMOI:

Finally, we have also included a comparison between naive offset MOI and MOIRE-estimated MOI and eMOI, respectively, in Supplementary Fig. 1).

“Estimates of multiplicity of infection (MOI) per sample ($N=382$) is compared between A. naive offset MOI, i.e., the second highest number of distinct alleles observed in one locus, and MOI estimated using MOIRE (R package), B. naive offset MOI, i.e., the second highest number of distinct alleles observed in one locus, and effective MOI (eMOI), i.e., adjusted for intra-host relatedness between parasite strains, also estimated with MOIRE, and C. MOI and eMOI, both computed with MOIRE.”

The formula for Fws is incorrect – it seems instead you have another formula for within-host heterozygosity? But this formula too I struggle to understand.

RESPONSE: *Thanks for pointing this out. Yes, the formula was for within-host heterozygosity. We apologize for this mistake. We have now corrected it and included the correct formula for 1-Fws. “1-Fws was calculated as the allele heterozygosity of the individual (H_w) relative to the population³⁷:*

$$1 - F_{ws} = \frac{H_w}{H_E}$$

We used 1-Fws in order to have increasing values with increasing diversity, aligned with the other metrics used. H_w is defined as:

$$H_w = 1 - \left(n_i \frac{1^2}{n_i} \right)$$

where n is the number of alleles detected at the ith locus of a given sample. Individual mean 1-Fws was calculated across all diversity loci.”

Line 77 and elsewhere, the phrase “Attempted sequenced” should just be “sequenced”; or maybe “selected for sequencing”.

RESPONSE: *Thanks for pointing this out. “Attempted” has now been deleted.*

Line 97, is this the same Fws statistic defined by Magnus Manske et al.

(<https://www.nature.com/articles/nature11174>)? If so, I think you should cite that manuscript.

Also, in personal experience, I’ve never heard this statistic described as “1 - Wright’s inbreeding coefficient”, rather just “Fws”, the latter also being more concise

RESPONSE: *Correct, it is the same statistic. We have included the Manske et al. reference, and substituted “Wright’s inbreeding coefficient” with “1-Fws” throughout the manuscript. We are using 1-Fws rather than Fws in order to have values that are positively correlated with diversity, similar to the other metrics we use in the analysis. We have included a sentence explaining this in the Methods (“We used 1-Fws in order to have increasing values with increasing diversity, aligned with the other metrics used.”).*

In Figure 1, please indicate in the figure caption what quantiles are demarcated by the box and whiskers in the boxplots

RESPONSE: *We have now included this information in the Figure 1 and 3 legends: “Boxes indicate the 25th and 75th percentiles, and the whiskers indicate the smallest value within 1.5 times interquartile range below the 25th percentile, and the largest value within 1.5 times the interquartile range above 75th percentile.”.*

In the caption of Figure 1B, could you please precisely describe what is meant by “covered”?

E.g. is a locus considered covered when it has a minimum of 1X coverage across all its bases?

RESPONSE: *This figure includes all allele data outputted from our bioinformatics pipeline before filtering by number of reads and coverage across samples and loci, so “covered” simply refers to having quality reads for our loci of interest. Thanks for pointing it out, we realize now that this can be misleading, and have changed the wording to “with reads”.*

The information in Figure 1D/E is important but the presentation could be improved. For example, in Figure 1D, the x-axis position of the amplicons has no indicated significance; rather just the label “Diversity”. If there is a significance, please indicate that. If not, the plot would be more informative if, for example, you ordered the amplicons by their median abundance, or the chromosome they are on, or both; anything that you think would help readers better appreciate the contents of your panel and/or the variation in sequencing performance across amplicons. Similarly, for Figure 1E, you could break “Resistance” by loci.

RESPONSE: *Thanks very much for this suggestion. We have now ordered and labeled the axis by chromosome for diversity markers, and genes for resistance markers, respectively.*

Figure 2 caption, states “eMOI in Magude in 0-truncated Poisson regression adjusted for parasitemia”, but p-values are shown for all three regions.

RESPONSE: *It applies to all three regions. Thanks for pointing out the mistake. “Magude” has now been deleted from the sentence.*

Figure 2 caption, “estimated with Poisson and logistic regressions similar to A-B”, should be A-C.

RESPONSE: *Correct. A-B has now been changed to A-C. Thanks for pointing it out.*

Throughout Population genetic diversity, the text and the figure panel labels do not match. For example, Fig. 3c,d is indicated when describing temporal trends; whereas this should be Fig. 3e.

RESPONSE: *We have fixed the mixing up of 3c,d and 3e,f in the text.*

Figure 3 caption and Line 359, when you randomly subsample, how many replicates do you perform? Also I would like to confirm this is random sampling without replacement.

RESPONSE: *The random subsampling refers to a single random draw of the population of larger size to balance it with the smaller population. This was done without replicates because of the computational expense of running MOIRE to obtain HE for each population. We have elaborated the sentence describing this in the Results section to make it clearer: “In order to compare HE between time windows, areas, and population groups, the larger populations were randomly subsampled within areas and/or years once without replacement to match the smaller*

population in size.”. We have also specified this in the legends of Fig.3 and Supplementary Tables 7 and 8, and in Methods - Statistical analysis. Thanks for pointing out the missing information.

Line 337, “Multiple Chain” should be “Markov Chain”

RESPONSE: We have now corrected the mistake.

Consider including the mathematical definition of eMOI in the Definitions section of the methods; personally, I found it much easier to understand than the current verbal description.

RESPONSE: Thanks for the suggestion, we agree that it aids the understanding with the mathematical formula, and it has now been included.

“Individual MOI and eMOI was also estimated with the MOIRE MCMC algorithm, accounting for allele frequencies across loci. eMOI furthermore takes within-host relatedness (r) into account, and can be interpreted as the expected MOI if population diversity was infinite ($H_E=1$). It is calculated as:

$$eMOI = 1 + (1 - r)(MOI - 1)$$

”

If eMOI takes continuous values, could you please comment on why you’ve chosen to do Poisson regression

RESPONSE: We chose Poisson regression because we wanted to respect the fact that eCOI can’t take negative values, and it is still sort of count values (count of distinct clones), although with decimals. We have now added the reasoning behind this choice to the statistical analysis section (“P-values and confidence intervals for eMOI were obtained from zero-truncated Poisson regressions in order to restrict eMOI to positive values.”).

Reviewers' Comments:

Reviewer #1:

Remarks to the Author:

The comments have been sufficiently addressed.

One final edit, that should be made clear in the text is the low numbers in Ilha Josina that precluded further analysis. As per the response given below.

Magude had a decline in eMOI and a comparison was done between ANC and children populations. Ilha Josina had an increasing trend over time in polyclonal infections, Fig 2a should be cited at the end of the sentence. However, why was no further analysis done on Ilha Josina?

RESPONSE: Unfortunately, the sample size for children in Ilha Josina was too small (N=4, after filtering) to do further comparisons between ANC users and children in this area. We would like to point out that there was no change over time in multiplicity of infection in Ilha Josina to support an increasing trend in polyclonality in the area, which also was not significant ($p=0.33$). Nonetheless, it would have been very interesting to compare ANC users in children in the other areas. We hope this study inspires future comparisons between ANC users and the community across different epidemiological settings.

Reviewer #2:

Remarks to the Author:

The authors have addressed all the concerns and suggestions in detail. The manuscript is easy to read and follow. The methods are explained in detail. The missing description and information have been included. The study design, data acquisition, analysis, statistical tests, significance, interpretation, and conclusions are in line with the expectations and results shown. The work meets expected standards in the field of malaria genomic surveillance.

Reviewer #3:

Remarks to the Author:

I appreciate the efforts the authors have made to address my comments.

In my first two major comments, I asked the authors to further explore and interpret the relationships they observe between intra-host genetic diversity and transmission intensity. The additional analysis, result section, and paragraph in the discussion do this well. Overall, the conclusion one draws from these additions is that the relationship between intrahost genetic diversity and transmission is, on the whole, quite weak / noisy / nuanced. While this is disappointing from a surveillance point of view, it is consistent with much of the existing empirical results in the literature (for e.g. <https://www.nature.com/articles/s41467-020-15779-8>, <https://www.nature.com/articles/s41467-023-43087-4>). I think the manuscript is much more valuable now that this is presented more explicitly, because it will both prevent naïve usage of genetic statistics for estimating transmission, and encourage the development of more sophisticated approaches going forward.

I have one remaining minor comment here: please adjust the language you use in the results section when discussing the PCC values. In particular, while all of the correlations fail to reach significance, several of the negative values (e.g. $PCC = -0.59$) would typically be described as moderate, not small. You could say, e.g. "small to moderate".

My third major comment about differences between ANC and children has been well addressed.

My forth major comment was about use of unpublished software and a novel statistic, eMOI, for your key results. Thank you for the additional analyses addressing this point – jointly, they have demonstrated very clearly that your results are robust to specifics about what MOI statistic is used, and strengthened the evidence supporting your conclusions.

One element of this new analysis I found surprising, however, and would appreciate if the authors would comment on. In the new SFig 1C, there is a comparison between MOI and eMOI, the latter incorporating within-sample IBD. This plot seems to show that MOIRE is estimating considerable within-sample IBD in nearly all infections. Prima facie, I would have expected density along the diagonal (representing super-infection, with very low IBD, given you have shown little interhost IBD in the manuscript) as well as samples with $eMOI \ll MOI$ (representing likely co-infection). Why is there so little on the diagonal? Also, my understanding of Dcifer, the tool you use to estimate interhost IBD, is that it assumes no within-sample IBD. I understand this may just be a limit of existing state-of-the-art tools, but can we be confident in the analysis with Dcifer given the considerable amount of the within-sample IBD found by MOIRE? Can you please clarify what effect this within-sample IBD have on the outputs or conclusion drawn from the Dcifer analysis?

Major comment five was about the Fws formula. Thank you for correcting this. However, your within-sample heterozygosity formula is still incorrect: in fact, it evaluates to zero. The correct formula will typically be taking an average of the within-sample heterozygosity across all your loci; where each locus you will compute one minus the squared frequency of the different alleles. I would suggest you check the code you used to calculate this, and provide the formula from there. I tried to check the code provided myself but it is just the source code for MOIRE (which I don't believe returns the heterozygosity).

All of my minor comments were nicely addressed, thank you.

REVIEWER COMMENTS

Reviewer #1 (Remarks to the Author):

The comments have been sufficiently addressed.

One final edit, that should be made clear in the text is the low numbers in Ilha Josina that precluded further analysis. As per the response given below.

Magude had a decline in eMOI and a comparison was done between ANC and children populations. Ilha Josina had an increasing trend over time in polyclonal infections, Fig 2a should be cited at the end of the sentence. However, why was no further analysis done on Ilha Josina?

RESPONSE: Unfortunately, the sample size for children in Ilha Josina was too small (N=4, after filtering) to do further comparisons between ANC users and children in this area. We would like to point out that there was no change over time in multiplicity of infection in Ilha Josina to support an increasing trend in polyclonality in the area, which also was not significant ($p=0.33$). Nonetheless, it would have been very interesting to compare ANC users in children in the other areas. We hope this study inspires future comparisons between ANC users and the community across different epidemiological settings.

RESPONSE: *Thank you for pointing out that we did not explain this directly in the text. We have now added the sentence "Further comparisons between ANC and children in the other two areas were precluded by limited number of samples from children." in the results section on temporal trends in within-host genetic diversity.*

Reviewer #2 (Remarks to the Author):

The authors have addressed all the concerns and suggestions in detail. The manuscript is easy to read and follow. The methods are explained in detail. The missing description and information have been included. The study design, data acquisition, analysis, statistical tests, significance, interpretation, and conclusions are in line with the expectations and results shown. The work meets expected standards in the field of malaria genomic surveillance.

RESPONSE: *Thank you again for the constructive comments and suggestions.*

Reviewer #3 (Remarks to the Author):

I appreciate the efforts the authors have made to address my comments.

In my first two major comments, I asked the authors to further explore and interpret the relationships they observe between intra-host genetic diversity and transmission intensity. The additional analysis, result section, and paragraph in the discussion do this well. Overall, the conclusion one draws from these additions is that the relationship between intrahost genetic diversity and transmission is, on the whole, quite weak / noisy / nuanced. While this is disappointing from a surveillance point of view, it is consistent with much of the existing empirical results in the literature (for

e.g. <https://www.nature.com/articles/s41467-020-15779-8>, <https://www.nature.com/articles/s41467-023-43087-4>). I think the manuscript is much more valuable now that this is presented more explicitly, because it will both prevent naïve usage of genetic statistics for estimating transmission, and encourage the development of more sophisticated approaches going forward.

RESPONSE: *Thank you for again for prompting us to make a more direct comparison with the epidemiological trends. We agree that it has improved the value of the manuscript.*

I have one remaining minor comment here: please adjust the language you use in the results section when discussing the PCC values. In particular, while all of the correlations fail to reach significance, several of the negative values (e.g. PCC = -0.59) would typically be described as moderate, not small. You could say, e.g. “small to moderate”.

RESPONSE: *We agree and have adjusted the statement to: “In Magude, eMOI in both ANC users and children showed positive Pearson’s correlation coefficients (PCC) close to 1 (>0.85), although not statistically significant. Furthermore, for ANC users in Magude, both $1-F_{WS}$ and proportion of infections that were polyclonal showed $PCC > 0.65$. In the other two areas, PCC was negative, but small to moderate and not statistically significant, for all three metrics of intra-host genetic diversity.”*

My third major comment about differences between ANC and children has been well addressed.

My fourth major comment was about use of unpublished software and a novel statistic, eMOI, for your key results. Thank you for the additional analyses addressing this point – jointly, they have demonstrated very clearly that your results are robust to specifics about what MOI statistic is used, and strengthened the evidence supporting your conclusions.

One element of this new analysis I found surprising, however, and would appreciate if the authors would comment on. In the new SFig 1C, there is a comparison between MOI and eMOI, the latter incorporating within-sample IBD. This plot seems to show that MOIRE is estimating considerable within-sample IBD in nearly all infections. Prima facie, I would have expected density along the diagonal (representing super-infection, with very low IBD, given you have shown little interhost IBD in the manuscript) as well as samples with $eMOI \ll MOI$ (representing likely co-infection). Why is there so little on the diagonal? Also, my understanding of Dcifer, the tool you use to estimate interhost IBD, is that it assumes no within-sample IBD. I understand this may just be a limit of existing state-of-the-art tools, but can we be confident in the analysis with Dcifer given the considerable amount of the within-sample IBD found by MOIRE? Can you please clarify what effect this within-sample IBD have on the outputs or conclusion drawn from the Dcifer analysis?

RESPONSE: *Thanks for bringing this up. It is important to distinguish between inter-host relatedness and intra-host relatedness, and we realize that it needs more clarification in the manuscript. Our interpretation of eMOI values being lower than MOI is that co-transmission is more common than super-infections in this setting, leading to inbreeding. We have now added this to the results section with “Effective multiplicity of infection (eMOI), which incorporates intra-host relatedness between clones, was lower*

than MOI at 1.8, indicative of co-transmission over super-infections, leading to inbreeding (Supplementary Fig. 1C).” Furthermore, in the discussion section, we have included that “Relatedness between clones within a sample was evident from eMOI being lower than MOI, indicating that co-transmission is a more common event than superinfections. For our main analysis, we take this relatedness into account by using eMOI rather than MOI to estimate within-host diversity.”

The low levels of inter-sample relatedness estimated with Dcifer suggests that there is little localized transmission. However, our ability to measure inter-sample relatedness is highly dependent on the sampling schedule. With our temporally sparse sampling of ANC users, we are probably not able to detect much localized transmission even if present. We have added a sentence in the discussion about this: “The very low inter-host relatedness observed at ANC might reflect little localized transmission, although more dense sampling would probably be required to detect it.”

We are confident that we can still trust the Dcifer results on inter-host relatedness even though we have substantial intra-host relatedness. In the Dcifer paper (<https://academic.oup.com/genetics/article/222/2/iyac126/6674513>), Gerlovina and colleagues test how different levels of intra-host relatedness affects the Dcifer inter-host relatedness estimates, and find no significant bias. They write that “Simulations with imposed intrahost relatedness indicate that the working model achieves its stated goal of capturing interhost relatedness by implicitly downweighting the independent contribution of related strains within a host to comparisons between hosts.”

Major comment five was about the Fws formula. Thank you for correcting this. However, your within-sample heterozygosity formula is still incorrect: in fact, it evaluates to zero. The correct formula will typically be taking an average of the within-sample heterozygosity across all your loci; where each locus you will compute one minus the squared frequency of the different alleles. I would suggest you check the code you used to calculate this, and provide the formula from there. I tried to check the code provided myself but it is just the source code for MOIRE (which I don't believe returns the heterozygosity).

RESPONSE: The formula was missing a parenthesis – thank you for spotting this. The correct one is:

$$H_w = 1 - \left(n_i \left(\frac{1}{n_i} \right)^2 \right)$$

To clarify, we first calculate the within-host heterozygosity per locus based on the number alleles observed, then calculate the $1-Fws = Hw/He$, and finally we take the average $1-Fws$ per individual across all loci. We have corrected the formula in the manuscript and hope that it is clear now.

All of my minor comments were nicely addressed, thank you.